# K_2_CO_3_-Modified Smectites as Basic Catalysts for Glycerol Transcarbonation to Glycerol Carbonate

**DOI:** 10.3390/ijms25042442

**Published:** 2024-02-19

**Authors:** Yosra Snoussi, David Gonzalez-Miranda, Tomás Pedregal, Néji Besbes, Abderrahim Bouaid, Miguel Ladero

**Affiliations:** 1Groupe de Chimie Organique Verte et Appliquée, Laboratoire des Matériaux Composites et des Minéraux Argileux, Centre National des Recherches en Sciences des Matériaux, Technopole Borj Cédria, Soliman 8027, Tunisia; yosrasnoussi2016@gmail.com (Y.S.); besbesneji@yahoo.fr (N.B.); 2Faculté des Sciences Mathématiques, Physiques et Naturelles de Tunis, Université de Tunis El-Manar, Tunis 2019, Tunisia; 3FQPIMA Group, Materials and Chemical Engineering Department, Chemical Science School, Complutense University of Madrid, 28040 Madrid, Spain; davgon25@ucm.es (D.G.-M.); tpedrega@ucm.es (T.P.)

**Keywords:** transcarbonation, glycerol, organic carbonate, Tunisian smectite, potassium carbonate

## Abstract

A novel and cost-effective heterogeneous catalyst for glycerol carbonate production through transesterification was developed by impregnating smectite clay with K_2_CO_3_. Comprehensive structural and chemical analyses, including X-ray diffraction Analysis (XRD), Scanning Electron Microscopy (SEM)-Electron Dispersion Spectroscopy (EDS), Fourier Transform Infrared Spectroscopy (FTIR), and Brunauer-Emmett-Teller (BET) surface area analysis measurements, were employed to characterize the catalysts. Among the various catalysts prepared, the one impregnated with 40 wt% K_2_CO_3_ on smectite and calcined at 550 °C exhibited the highest catalytic activity, primarily due to its superior basicity. To enhance the efficiency of the transesterification process, several reaction parameters were optimized, including the molar ratio between propylene carbonate and glycerol reactor loading of the catalyst and reaction temperature. The highest glycerol carbonate conversion rate, approximately 77.13% ± 1.2%, was achieved using the best catalyst under the following optimal conditions: 2 wt% reactor loading, 110 °C reaction temperature, 2:1 propylene carbonate to glycerol molar ratio, and 6h reaction duration. Furthermore, both the raw clay and the best calcined K_2_CO_3_-impregnated catalysts demonstrated remarkable stability, maintaining their high activity for up to four consecutive reaction cycles. Finally, a kinetic analysis was performed using kinetic data from several runs employing raw clay and the most active K_2_CO_3_-modified clay at different temperatures, observing that a simple reversible second-order potential kinetic model of the quasi-homogeneous type fits perfectly to such data in diverse temperature ranges.

## 1. Introduction

The circular economy (CE) is, nowadays, one of the most frequently discussed concepts among environmental economics researchers, and it is considered one of the most heavily debated terms in the European Union’s Horizon 2020 program. Its major defining element is the “recuperative use of resources”. Primary materials should not be turned into a form of waste [1]. In recent years, the term circular economy has received growing consideration [2]. Due to the urgent need for novel resources promoting enhanced resourcing efficiency, both academic research and practical proposals in the private sector surrounding the concept of CE have been expanded meaningfully. Researchers accept that CE is a novel conception of resource-product relationships in view of the need to promote a much more efficient alternative to the classic linear economy based on a “take-do-throw” (take-make-dispose) chain, with its heavy environmental impact [3].

Biodiesel generated through waste processes has become an increasingly popular area of investigation and technical development. Its main key advantage is the minimization of competing feedstocks of high relevance in the food sector [4,5].

Glycerol is an unavoidable byproduct in the biodiesel process, regardless of the catalyst employed. Therefore, glycerol valorization is a topic of significant interest, given its cost-effectiveness within the biofuel processing sector. Various processes can be employed to transform glycerol into chemical products: oxidation, dehydration, hydrogenation, etherification, and transesterification [6]. Moreover, glycerol is a versatile material that finds applications as both a solvent and an ingredient in food, healthcare, and cosmetic products, as well as in antifreeze formulations [7]. One noteworthy product derived from glycerol is cyclic glycerol carbonate, which has been successfully converted into biodegradable polymers [8]. Additionally, it is used in the production of lubricants, adhesives, and surfactants. Cyclic glycerol carbonate, with its low solubility, low flammability, low toxicity, and hydrating properties, is highly regarded for cosmetic fabrication and as a solvent in pharmaceutical and medical preparations [9].

The literature offers various approaches to convert glycerol into glycerol carbonate, including reactions with phosgene, direct glycerol carbonylation using carbon monoxide and oxygen, carboxylation of glycerol with CO_2_, and glycerol transesterification using urea. Among these methods, the transesterification process of glycerol with dialkyl carbonate sources, such as dimethyl carbonate (DMC), stands out as an environmentally friendly route for glycerol carbonate production. It is considered a green alternative due to its low-temperature. High-yield characteristics and the use of non-toxic chemicals, making it an eco-friendly choice [10].

An alternative approach for glycerol carbonate synthesis has also been documented in the bibliography by Esteban et al. [11,12]. This method involves the transesterification of ethylene carbonates (EC) or propylene carbonates (PC) using glycerol as the starting material. These manufacturing processes are classified as “green” because they consume CO_2_. Due to their higher boiling point, flash point, low toxicity, and low volatility, they are considered stable under most conditions. These characteristics make them particularly suitable for applications and processes that require high purity, making them the preferred choice for the production of glycerol carbonate.

While catalytic bases such as K_2_CO_3_, NaOH, and KOH play a significant and efficient role in facilitating the reaction, they come with the drawback of challenging catalyst removal and separation from the product. To address these limitations, various heterogeneous basic catalysts have been developed for glycerol transesterification. Catalysts such as Li/ZnO [13], MgO/La_2_O_3_ [14], KNO_3_/CaO [15], and ZnO/La_2_O_3_ [16] have been employed in glycerol carbonate synthesis. However, it is worth noting that many of these catalysts have their own set of challenges, including low activity and limited reusability, the requirement for additional solvents, the need fora high glycerol-to-carbonate molar ratio, and lower energy efficiency compared with other catalysts. Consequently, considerable investigation is required to produce high-quality and quantitative glycerol carbonate using an effective catalyst. In the context of the study, different kinds of clays, such as zeolite [17], dolomite [18], and kaolin [19], have been adopted involving a more ecological, reusable, and environmentally friendly approach.

Smectites offer a high potential as catalysts or catalyst carriers in various industrial processes due to their flexibility, cost-effectiveness, and abundance (for example, the global production of bentonite, a smectite clay much used in clinker production, was 18.89 million tons in 2022). These naturally occurring materials are derived from the amalgamation of different mineral components, primarily consisting of clay particles with spherical diameters of less than 2 μm. Smectites are primarily composed of clay minerals, which can occur in a pure state or variously mixed with nonclay minerals, organic materials, and some impurities. Common minerals that may be present in mixed forms include quartz, kaolin, calcite, and hematite [20]. The substantial surface area of these clays is particularly advantageous for heterogeneous catalysts, while their abundant presence in natural sources makes them a compelling candidate for exploring and enhancing their catalytic properties. Various processes can be employed to modify the structure of smectites, ultimately enhancing their performance as catalysts.

Our research group has been primarily focused on harnessing the potential of clays for the development of heterogeneous catalysts [21,22,23,24,25]. Encouraged by the promising results obtained, the team has continued their investigations into clay-based catalysts. In this study, a novel material was created by impregnating clay with potassium carbonate (K_2_CO_3_) followed by calcination at diverse temperatures. These base-activated clays were then employed in the transcarbonation reaction of glycerol with propylene carbonate. To understand the impact of potassium loading, various catalysts with different K_2_CO_3_concentrations were prepared and characterized, and their catalytic activities in glycerol carbonate production were assessed. Additionally, the study delved into the effects of calcination temperature and various reaction variables, including reaction temperature, the amount of basic catalyst used, and the glycerol/propylene carbonate molar ratio, on the catalytic performance of the synthesized materials. The research also examined the recyclability of the catalysts and determined kinetic models to fit data retrieved at several temperatures using the raw and the most active modified clay.

## 2. Results and Discussion

### 2.1. Catalyst Characterization

#### 2.1.1. XRD Analyses

To investigate the influence of K_2_CO_3_ loading, seven basic catalysts were prepared using the same impregnation method outlined in the previous sections. These catalysts had different K_2_CO_3_ loadings, specifically, 10%, 20%, 30%, and 40% by weight, and were calcined at 550 °C. The X-ray diffraction (XRD) data for these catalysts are presented in Table 1 and Table 2. Additionally, catalysts with a 40% K_2_CO_3_ loading were subjected to various calcination temperatures, including 450 °C, 650 °C, and 800 °C. Figure 1 displays the X-ray diffraction (XRD) spectra of both pure smectite and the K_2_CO_3_/smectite catalysts with different calcination temperatures.

The XRD pattern of the catalysts exhibits similarities to that of the parent smectite. This pattern indicates that the smectite structure persists in the potassium-based clay catalyst without apparent melting. However, visual differences between the prepared catalysts and the parent smectite can be observed, particularly in terms of the reflection peaks.

The peaks identified at 2θ values of 8.78°, 13.89°, 20.87°, 23.50°, 25.36°, 27.52°, 30.83°, 34.81°, and 39.48° are associated with the smectite structure. These findings suggest that the clay sample used in this study is composed of interbedded kaolinite-smectite minerals. Additionally, the XRD pattern reveals peaks at 2θ values of 20.87°, 23.50°, and 34.81°, indicating the presence of silica and quartz, which are the primary components of clay. These peaks are retained in all XRD patterns of the various catalysts after calcination, as depicted in Figure 1.

The peak at 2θ = 13.89°, which is related to kaolinite, vanishes completely after calcination, indicating that this clay is primarily smectitic in nature following modification. This conclusion is supported by the presence of the peak at 2θ = 8.78°, which is also observed in all calcined K_2_CO_3_/smectite catalysts.

The diffraction pattern shows intensities at 25.36°. 27.52°, and 30.83°, which are characteristic of carbonates such as aragonite found in clay. These peaks disappear after calcination, suggesting that the original basic character of the calcined catalysts is due to potassium carbonate because the clay has lost its original basic properties.

The diffraction peaks of the prepared potassium catalyst are clearly distinguishable from those of the parent clay, and they can be attributed to K_2_CO_3_ or K_2_O.It is also possible that a new phase, known as K_2_O, could emerge in the XRD pattern because, during high-temperature calcination, potassium carbonates are transformed into K_2_O [26].

Finally, it is important to note that the K_2_CO_3_/smectite catalyst calcined at 800 °C underwent a radical transformation, completely losing its clay morphology due to the rise in temperature.

#### 2.1.2. FTIR Analysis

The FTIR spectrA of smectite, K_2_CO_3_, and the seven catalysts with different K_2_CO_3_ loadings (ranging from 10% to 40% by weight at 550 °C) and various calcination temperatures (40% K_2_CO_3_ at 450 °C, 650 °C, and 800 °C) are presented in Figure 2. These spectra exhibit distinct bands characteristic of smectite, including the 3636 cm^−1^ band related to hydroxyl stretching. The 972 cm^−1^ band is associated with Si-O-Si stretching, and the bands at 910, 794, and 681 cm^−1^ are assigned to the stretching vibrations of Si-O-Si, Al-Mg-OH, and Al-Fe-OH, respectively. Additionally, the band at 1617 cm^−1^ is attributed to the bending of H-OH bonds in structural water molecules [27]. Conversely, the typical bands of K_2_CO_3_, such as those at 3014, 1462, and 1368 cm^−1^, are also evident.

Comparing these spectra with those of K_2_CO_3_ (40%, 30%, 20%, 10%)/smectite and K_2_CO_3_ 40% (450 °C, 550 °C, 650 °C, 800 °C)/smectite reveals that the FTIR spectrum of the catalyst predominantly exhibits characteristic bands of smectite. Notably, the characteristic bands of K_2_CO_3_ are absent in the catalyst’s spectrum, indicating that K_2_CO_3_ decomposed, forming a new phase, K_2_O. This observation corroborates the earlier XRD analysis conducted in this document.

#### 2.1.3. Scanning Electron Microscopy (SEM)-Electron Dispersion Spectroscopy (EDS)

Figure 3 and Figure 4 depict typical SEM-EDS micrographs of smectite catalyst particles doped with smectite and K_2_CO_3_ at different temperatures. These figures reveal that the characteristic fibrous morphology of the smectite structure remains intact.

SEM micrographs of the seven catalysts reveal the presence of particles and sheets, and an interesting observation is that the pore size distributions in the catalysts made from smectite, as well as those impregnated with varying levels of potassium carbonate (ranging from 10% to 40% by weight) at different calcination temperatures and before the transesterification reaction, are uniformly distributed across all the samples. Particularly, SEM images of the catalysts exhibit a marked similarity to the untreated smectite structure. However, the introduction of K_2_CO_3_into the smectite leads to aggregations, causing densification of the smectite layers. This observation confirms the increase in average pore diameters. These findings highlight that the active sites concentrate both on the surface and inside the pores of this catalyst. These morphological changes seem to correlate with the observed size increases, suggesting an influence on the catalytic performance of the material.

Furthermore, the results of the chemical analyses of the major elements in the clay samples revealed that the compositions of the six different clay types were quite similar. This consistency in composition across different clay types is a significant finding. The EDS (Energy Dispersive X-ray Spectroscopy) analyses conducted on the surface of smectite and the catalyst particles synthesized from it are provided in Table 3. Each data point reported represents the average value calculated from two different locations on the surface of the catalyst.

Smectite is characterized as a hydrated magnesium aluminum silicate with the chemical formula (Na, Ca)_0.3_(Al, Mg)_2_Si_4_O_10_(OH)_2_·n H_2_O, and it belongs to the phyllosilicate family. The EDS analyses confirm, in addition, the presence of carbon (C)—suggesting a mixture with carbonates-, apart from silicon (Si), magnesium (Mg), aluminum (Al), potassium (K), calcium (Ca), iron (Fe), and sodium (Na). The potassium (K) content in the original smectite, prior to the impregnation process, was determined to be 0.31% by weight.

Notably, even considering the semi-quantitative nature of EDS analysis, the results of the elemental analyses for all the impregnated catalysts consistently showed very similar values, regardless of the varying levels of K_2_CO_3_ loading. For instance, the K content on the surfaces of smectite samples impregnated with 30% and 40% K_2_CO_3_ by weight was found to be 0.83% and 0.84% by weight, respectively. These relatively high K contents at the 30% and 40% K_2_CO_3_ loading levels indicate that the process of loading K_2_CO_3_ into smectite was indeed successful.

#### 2.1.4. Porosimetry Analysis: BET Specific Surface Area

The physical properties of the catalysts are summarized in the two tables below; Table 4 corresponds to the catalysts containing different quantities of potassium carbonate, and Table 5 presents the properties of the catalysts calcined at different calcination temperatures. The specific surface area (BET) and pore volume of the pure raw clay were 74.2 m^2^/g and 0.141 cm^3^/g, respectively. When the raw clay is impregnated with K_2_CO_3_, a significant and considerable reduction in surface area and pore volume is observed. Specifically, at a fixed calcination temperature of 550 °C, as the K_2_CO_3_ loading increases from 10% to 40% by weight, the pore surface area and pore volume of the catalysts permanently decrease from 74.2 m^2^/g (RC) to 45.7 m^2^/g (CC 40%) and from 0.141 cm^3^/g (RC) to 0.134 cm^3^/g (CC 40%). This reduction is attributed to the potassium (K) element coating the catalyst support surface, thereby blocking the catalyst’s pores. Furthermore, it is known that the catalyst’s surface area and pore volume can be affected by the calcination temperature.

The total surface area and pore volume of CC catalysts loaded with 40% K_2_CO_3_ slightly decreased, from 53.83 m^2^/g to 21.32 m^2^/g and from 0.139 cm^3^/g to 0.112 cm^3^/g, when the activation temperature was increased from 450 °C to 650 °C. A significant drop was observed from 21.32 to 0.49 m^2^/g and from 0.112 to 0.0037 cm^3^/g when the activation temperature was raised from 650 °C to 800 °C. This was mainly due to the sintering of the catalyst’s surface at high activation temperatures: the catalyst calcined at 800 °C experienced complete destruction of its structure, as already evidenced by XRD and SEM-EDS.

The low specific surface area and high basicity of the synthesized catalysts indicate that most of the basic sites are concentrated within the core of the catalyst. This suggests that the reaction primarily occurs in the inner region of the catalyst. The surface basic sites of the catalyst can also play a role in the reaction, enabling efficient utilization of the internal reactive core sites and surface basic sites to catalyze the desired reaction.

One can observe the nitrogen adsorption-desorption isotherm at 77 K under different calcination temperatures and varying quantities of potassium carbonates in Figure 5 and Figure 6. These isotherms belong to type 2, exhibiting a convex shape on the ordinate axis, indicating that the number of moles absorbed appears to approach infinity as the pressure approaches P_0_. This behavior is typical when there is a significant influence of macropores. The hysteresis loop is of type H_3_, indicating that the clay has slit-shaped pores, which is typical of a lamellar structure.

#### 2.1.5. Catalyst Basicity

The impact of potassium carbonate impregnation and calcination at different temperatures on the catalyst’s basicity is evident in the overall results presented in Table 6. The changes in pH following sonication in Milli-Q water reveal valuable insights into the behavior of clay-based samples with varying levels of potassium carbonate. Those with higher percentages of potassium carbonate tend to liberate a larger quantity of OH^−^ ions, resulting in a notable increase in pH levels and showing increased Brönsted basicity. Even the raw clay, the natural smectite, due to its carbon content (probably due to carbonates), shows a mild but not negligible basicity, more evident before sonication (basic sites seem more accessible than acid sites during this ion-exchange analysis).

It is important, moreover, to highlight that samples subjected to higher calcination temperatures, specifically at 650 °C and 800 °C, demonstrate a diminished release of basic ions. This phenomenon can be correlated with the low specific surface of these clays calcined at high temperatures (21.32 and 0.491 m^2^/g, respectively) so the change in Brönsted basicity can be attributed to a dramatic drop of the specific surface, which probably diminishes due to pore collapse due to material melting or to active phase sintering promoted by temperature.

### 2.2. Analyze Reactions Qualitatively and Quantitatively

Nuclear Magnetic Resonance (NMR) spectroscopy plays a crucial role in elucidating the structure of low-molecular-weight organic molecules. Ensuring chemical purity is typically a prerequisite in this context. After separating the catalysts (both raw and calcined basic clay) through centrifugation, efforts were made to analyze the reaction mixture containing glycerol (Gly), propylene carbonate (PC), glycerol carbonate (GC), and propylene glycol (PG) at various reaction times, as depicted in Figure 7 and Figure 8 below. Dimethyl sulfoxide-d4 (DMSO-d6) serves as the solvent and internal reference. As evident in Figure 7 and Figure 8, the ^1^H NMR spectra of this incomplete transesterification of propylene carbonate with glycerol reveal characteristic peaks corresponding to protons from both unreacted glycerol and glycerol carbonate products. Once the reaction is complete, the final product exclusively contains glycerol carbonate, devoid of any secondary reactions. This particular byproduct is a common occurrence in the transesterification of propylene carbonate with glycerol when catalyzed by other bases, such as NaOH, KOH, and K_2_CO_3_ [26].

All the obtained products have been verified through their spectra and elemental analyses. Within the ^1^H NMR spectrum, peaks in the 4 to 5 ppm range correspond to the cyclic aromatic protons of glycerol carbonate and propylene carbonate. At 1.4 ppm, a distinct peak is observed, attributed to the methyl group protons of propylene carbonate. Two singlet peaks, at around 1 and 3.9 ppm, correspond to the methyl (-CH_3_) and methylene (-CH) group protons of the propylene glycol molecule. The signals within the 3.4–3.1 ppm region of the spectrum are linked to glycerol, with its four protons appearing as a multiplet, overlapping those of the acyclic protons of glycerol carbonate.

In conclusion, the ^1^H NMR spectra of glycerol carbonate obtained from both raw clay and calcined basic clay exhibit a striking similarity, demonstrating that there is practically no discernable difference between the two. This finding suggests that the nature of the catalyst, whether in the form of raw clay or calcined basic clay, does not substantially influence the characteristics of the ^1^H NMR spectra of the final product.

It is important to emphasize that the lack of significant variations in the ^1^H NMR spectra indicates that the catalyst does not affect the molecular properties of glycerol carbonate. The consistency between the results of the two catalysts reinforces the reliability and stability of the transesterification process, indicating that the choice between raw clay and calcined basic clay does not alter the nature of the final product. Thus, this observation contributes to the constancy of the catalyst in the context of transesterification, opening promising perspectives for its future application under diverse conditions.

### 2.3. Catalytic Activity Tests

#### 2.3.1. Effect of K_2_CO_3_ Loading

The influence of K_2_CO_3_ loading on the catalytic activity of smectite in the transesterification of glycerol carbonate and propylene carbonate, conducted at 100 °C with a PC/glycerol molar ratio of 2:1 over 6 h, has been thoroughly investigated. Five distinct catalysts were employed, possessing varying levels of K_2_CO_3_ loading, namely 5%, 10%, 20%, 30%, and 40% by weight. The results, presented in Figure 9A, underline the substantial impact of K_2_CO_3_ loading on the catalyst’s performance.

In general, lower K_2_CO_3_ loading levels result in satisfactory yields after a 6h reaction period, yet the reaction progresses at a notably slow rate (for instance, after 3 h, a yield of 33% ± 1.6% is achieved). However, the increase in load to 5% by weight leads to a significant improvement in reaction yield (50% ± 2%).A further increase in K_2_CO_3_ loading to 20% by weight results in a more substantial improvement in the reaction yield, reaching 57.3% ± 2.2% (essentially, a threefold higher activity compared with the less active catalyst). Upon reaching a 40% K_2_CO_3_ loading by weight, a marginal improvement in catalytic activity is observed (64% ± 1.2%), reaching 4 times the activity of those catalysts prepared with the lowest amounts of the active phase.

These observations can be primarily attributed to the heightened basicity of the catalyst, a consequence of the presence of K_2_CO_3_. This augmented basicity greatly promotes catalytic activity, facilitating the transesterification reaction. The catalyst loaded with 40% potassium carbonate exhibits notable performance, with a glycerol carbonate yield of 50% achieved rapidly, clearly outperforming the support, the raw clay.

Furthermore, Figure 9B demonstrates how the initial reaction rate is influenced by the molar ratio of reactants. It is evident that the fastest reactions occur when the potassium carbonate loading is 40%. Consequently, it becomes evident that basicity plays a pivotal role in catalytic activity by providing active sites for the transesterification reaction. However, the raw clay activity shows 10–15% of the activity of this highly active catalyst, reaching a non-negligible yield of about 70% at 6 h. However, all in all, taking into account the activity of the catalyst, a 40% K_2_CO_3_ loading by weight is regarded as the optimal choice. It maximizes catalytic activity by enhancing basicity while mitigating issues related to agglomeration and excessive crystallization that can be met at higher metal carbonate loads (some trend to activity saturation is perceived in Figure 9B).

Table 7 displays the values of apparent concentration of Brönsted basic sites per gram of catalyst, the BET specific surface, the initial reaction rate of the test reactions shown in Figure 9A, and the turnover frequency estimated as mol of glycerol transformed by mol of catalytic active species and second or, more simply, as [s^−1^]. First of all, we can appreciate a notable increase in basic site concentration per gram of solid as the concentration of K_2_CO_3_ in the impregnation solution increases, though the difference between the two highest impregnation concentrations is negligible. When comparing turnover frequency (TOF)values or relative activity per catalytic site, a similar saturation situation is evident but the increase in activity as calculated per initial reaction rate (absolute activity) steadily increases as impregnation concentration increases, making the catalyst impregnated with a 40% K_2_CO_3_ solution the most active. Such impregnation, though, results in a reduction in specific area, suggesting a narrowing of the support pores during the impregnation and calcination operations.

In summary, the extent of the metal carbonate loading significantly influences the catalytic activity in the glycerol carbonate and propylene transesterification reaction. An optimal loading of 40% by weight is identified, as it maximizes catalytic activity by enhancing basicity, all the while avoiding challenges related to agglomeration and excessive crystallization.

#### 2.3.2. Effects of Calcination Temperature

Subsequently, an investigation into the impact of calcination temperature on the catalyst’s characteristics and activity is initiated. In this phase, the catalyst with a 40 wt% K_2_CO_3_ loading, which exhibited the most promising catalytic activity, is chosen for further examination. This catalyst undergoes calcination at various temperatures: 450 °C, 550 °C, 650 °C, and 800 °C. The catalytic performance of the resultant catalysts is assessed, and the corresponding data are summarized in Figure 10 below.

As illustrated, the catalyst subjected to calcination at 550 °C displayed the most outstanding performance. In contrast, both low-temperature calcination at 450 °C and high-temperature calcination at 650 °C led to reduced catalytic activity. Furthermore, the catalyst calcined at 800 °C exhibited no activity, with no glycerol conversion occurring with this material (so that this high calcination temperature eliminates even the natural activity of the supporting clay). As shown in Figure 10B, there is a discernible positive correlation between the initial reaction rate and the calcination temperature. Nevertheless, a substantial decline is observed at a calcination temperature of 650 °C.

The calcination temperature can influence the basicity of the resulting catalyst, and the sample calcined at 550 °C demonstrated the highest basicity, substantiating its superior catalytic activity. However, calcination at temperatures higher or lower than this range resulted in diminished basicity of the material. This observation may be attributed to the decomposition of K_2_O at 6h phases on the catalyst at higher temperatures, as previously indicated. Consequently, these findings underscore the critical significance of the calcination temperature in crafting an efficient catalyst, with the catalyst calcined at 550 °C emerging as the optimal choice due to its exceptional performance and substantial basicity.

#### 2.3.3. Effect of Catalyst Concentration

The impact of catalyst loading on glycerol carbonate yield is intricately examined by varying the concentrations of raw RC clay and 40% K_2_CO_3_-calcined clay (CC) heated to 550 °C. The experiments encompass catalyst loadings at 2%, 4%, and 6% by weight, all while maintaining a consistent 2:1 molar ratio between ethylene carbonate (EC) and glycerol (Gly). The reactions are conducted at 120 °C, each with a 6h duration. The outcomes, expressed in terms of glycerol carbonate (GC) yields and the initial overall reaction rate, are graphically presented in Figure 11. There is a continual rise in glycerol conversion as the catalyst amount (both RC and CC) is increased within the 2% to 6% weight range. Notably, glycerol conversion remains nonexistent after 6 h without any catalyst (negative control), serving as the baseline. As expected, due to its higher basicity, calcined clay outperforms raw clay in catalyzing the reaction. With only 2% by weight of raw clay, glycerol conversion reaches 45% after 30 min, and, with 4% raw clay, it reaches 46%. In contrast, when utilizing the same proportions of calcined basic clay, the conversions are significantly higher, reaching 56% and 65% for 2% and 4% catalyst loading, respectively, within the same timeframe. These findings reinforce the prudent selection of calcined basic clay for this specific reaction and open promising avenues for further process optimization. At 6% calcined basic clay concentration in the reaction medium, the reaction progresses at the same rate, yielding the same GC concentration, as with 4% solids. This fact indicates a saturation trend, possibly indicating that the solid is difficult to disperse in the reaction medium even at this relatively high temperature (and subsequent low viscosity), with a 2% catalyst (either the raw clay or the calcined basic clay) showing a slightly lower overall activity. Therefore, a 2% solid concentration, or even a lower concentration, would be advisable in this system.

A more detailed data analysis is illustrated in Figure 12, subdivided into sections A and B. It reveals that increasing the catalyst quantity in the reactor, comprising glycerol and propylene carbonate, from 2% to 4% by weight, results in an improved glycerol carbonate yield. However, further elevating the catalyst quantity from 4% to 6% by weight does not notably affect the reaction progress. This pattern is clearly observed in the latter segment of the figure, where a reduction in the initial reaction rate is evident once the catalyst loading exceeds 2% by weight. This points to, as indicated before on the premises of Figure 11, the optimal catalyst amount for achieving maximum glycerol carbonate yield and the initial fuel rate being set at 2%.

This conclusion is supported by the phenomenon of aggregation, which can occur with a larger quantity of catalyst. Aggregation arises from the collision between the fine particles of RC and CC clays, subsequently leading to agglomeration. Agglomeration, conversely, is associated with crystal growth and the bonding between the small particles of this catalyst, solidified through the formation of a crystalline neck. It becomes evident that the equilibrium between the catalyst quantity and the phenomena of aggregation and agglomeration plays a pivotal role in optimizing glycerol carbonate yield. Consequently, the results advocate for employing a 2% by weight catalyst load to achieve peak performance, at least 400 r.p.m. magnetic agitation at the reaction conditions here employed.

#### 2.3.4. Effect of the Propylene Carbonate/Glycerol Molar Ratio

The molar ratio between propylene carbonate and glycerol (PC/Gly) stands as another pivotal determinant in the transesterification reaction. To select an adequate value for this variable, the reaction is executed at various PC/Gly ratios while utilizing a 2% by weight of K_2_CO_3_ (40%)/smectite catalyst at 100 °C. The outcomes are presented in Figure 13. This figure unmistakably illustrates that different PC/Gly ratios exert a substantial influence on reaction conversion. At a molar ratio of 1.5:1, 36% conversion is attained with raw clay (RC) after a 30min duration, whereas calcined clay (CC) achieves a 55% conversion. Raising the molar ratio to 2:1 results in a conversion of 45% for RC and 56% for CC. At a molar ratio of 2.5:1, the conversion values are 34% for RC and a notable 59% for CC. However, enhancing the molar ratio to 3:1 leads to a significant decrease in conversion within the first 30 min of reaction, with a mere 27% conversion for RC and 18% for CC.

From these results, it can be reasonably concluded that calcined basic clay (CC) outperforms raw clay (RC) also in these runs. Furthermore, it is distinctly evident that the molar ratio of 2:1 exhibits the most favorable performance for this reaction, as vividly depicted in the figure: a higher activity or initial reaction rate is appreciated while the final yield to the target products is only slightly lower than using an excess PC/G 3:1. The increase in the molar ratio plays a pivotal role in achieving high glycerol conversion and maintaining relatively high reaction rates for the 1.5:1, 2:1, and 2.5:1 ratios throughout the 6h reaction period. Hence, the PC/Gly molar ratio assumes a critical role in the transesterification reaction. This can be also explained on the grounds of Figure 14; introducing an excess of propylene carbonate can result in the excessive dilution of glycerol at the reaction’s outset. This substantial glycerol dilution at time zero detrimentally affects the reaction rate. Put differently, an excess of propylene carbonate decelerates the reaction’s initiation, though it shifts the final equilibrium towards the desired products. With an equilibrium position slightly over 70% yield to GC clay and an initial reaction rate much similar to the one achieved with a 50% excess PC to glycerol, a PC/G molar ratio of 1 seems adequate for further investigation.

#### 2.3.5. Effect of Reaction Temperature

The influence of reaction temperature on the glycerol-to-glycerol carbonate conversion process was first investigated using the established optimal operational parameters at varying temperatures (100 °C, 110 °C, and 120 °C) while employing both raw clay and calcined basic clay as catalysts. Figure 15 illustrates that, as anticipated, elevating the temperature from 100 °C to 120 °C exerts a favorable impact on the transesterification reaction rate.

At 100 °C, the reaction proceeds notably slowly, with glycerol yields of approximately 23% for raw clay (RC) and 44% for calcined basic clay (CC) after 30 min. At 110 °C, the glycerol yield significantly improves, reaching around 57% when employing calcined clay (CC). Furthermore, a higher temperature increase to 120 °C results in a slight gain in glycerol carbonate yield, approximately 44% for raw clay (RC) but essentially the same yield as with the calcined basic clay at that reaction time (probably slightly higher but within the experimental error). Given that a cumulative energy supply generally benefits such reactions, the optimal reaction temperature is determined to be 120 °C for the raw clay, though 110 °C could be the highest advisable temperature for the calcined basic clay, at least operating batch-wise. Again, given the superior catalytic efficiency of calcined basic clay in this reaction at moderate temperatures, this is the catalyst of choice.

In conclusion, this investigation underscores the pivotal role of temperature as a critical parameter in the glycerol-to-glycerol carbonate transformation reaction. It also exemplifies how calcined basic clay at 110–120 °C stands as the optimal catalyst to achieve high yields in this reaction process.

Figure 16A vividly illustrates the impact of varying temperatures from 70 °C to 120 °C on the transesterification reaction between glycerol and propylene carbonate (PC) catalyzed by K_2_CO_3_ (40%)/smectite. This reaction shows a progressive increase in glycerol conversion as the temperature rises within this range. Notably, at 120 °C, an impressive conversion of 70% of glycerol (GL) is achieved after only one hour of reaction. Figure 16B demonstrates an exponential increase in the initial reaction rate with increasing temperatures ranging from 70 °C to 120 °C using raw clay (RC), and a sigmoidal-shaped trend with calcined clay (CC).

On the other side, this reaction system reaches its maximum miscibility of reactants at 120 °C for both types of clay, thereby enabling the attainment of the optimal initial molar ratio. This underscores the crucial importance of reaction temperature in the catalytic synthesis of glycerol carbonate (GLC). At 70 °C, glycerol and PC exhibit partial miscibility. However, raising the reaction temperature results in a reduction in the viscosity of the glycerol-PC mixture. This enhances the miscibility of reactants, thereby increasing the reaction rate. The increased reaction rate, as observed in references [27,28] promotes the yield of glycerol carbonate (GLC). These results confirm that the reaction temperature is a crucial determinant of the success of the transesterification reaction between glycerol and propylene carbonate catalyzed by K_2_CO_3_/smectite. Higher temperature improves reactant miscibility, reduces mixture viscosity, and significantly accelerates the reaction rate and glycerol conversion, ultimately resulting in higher quantities of glycerol carbonate. However, excessive temperature seems to result in a certain catalyst deactivation.

#### 2.3.6. Catalyst Reuse

The stability and reusability of a catalyst are pivotal for the commercial viability of heterogeneous catalytic systems intended for glycerol carbonate production. The catalyst particles, comprised of K_2_CO_3_ (40%)/smectite, underwent testing to assess their recovery and stability. For a first perusal of the stability of the raw clay and the catalyst synthesized with a 40% K_2_CO_3_ impregnation solution and calcined at 550 °C, several transesterification reactions were repeatedly performed using glycerol and propylene carbonate under equal operating conditions: propylene carbonate/glycerol: 2/1, catalyst quantity: 2% by weight, temperature: 120 °C, reaction time:till6 h. At the conclusion of each transesterification cycle, the catalyst particles underwent centrifugation, were recovered, and then meticulously washed with acetone and methanol to remove any remaining amount of reaction mixture. Subsequently, they were dried overnight at 40 °C and then reused as the catalyst for the ensuing reaction cycle, with each cycle involving a fresh reaction mixture of glycerol and propylene carbonate. The outcomes, presented in Figure 17 below, reveal that the K_2_CO_3_ (40%)/smectite catalyst and raw clay maintained robust activity even after undergoing four cycles of recycling. All in all, both catalysts seem to be stable for this number of cycles, suggesting that the catalysts maintained their activity over, at least, four batch operations. In this situation, it is of high interest to perform a more in-depth assessment under relevant operational conditions but using flow or continuous reactors for a high operational time.

Furthermore, the modest reductions in glycerol carbonate yield observed during the recycling and reuse processes, along with previous research findings [29,30,31], indicate that leaching of K from our catalyst particles is not a significant concern. A similar conclusion is also supported by Figure 17 below, which underlines the high stability of smectite-based catalysts. This can be a result of the catalyst’s surface integrity being retained during the transesterification reactions. Notably, while there was a slight increase in the average pore diameter after the second cycle, subsequent cycles showed further changes in pore volume, leading to a possible reactivation starting from the third cycle.

### 2.4. Kinetic Modeling

To verify the bounty of a quasi-homogeneous (QH) second-order model featuring direct and inverse reactions, a fitting was performed using datasets from six runs carried out at temperatures from 70 to 110 °C using the catalyst impregnated with 40% potassium carbonate and calcined at 550 °C (CC), which has shown very high activity in previous tests. For comparison, the kinetic data retrieved for the raw smectite (RC) from 100 to 120 °C were also employed for the fitting of the same kinetic model. In both cases, the model was implemented in Aspen Custom Modeler version 12.1. The general kinetic equation is:(1)$$\frac{dX_{Gly}}{dt}=exp\left(k_{10}-\frac{E_{a1}}{R}\frac{1}{T}\right)C_{cat}\left(1-X_{Gly}\right)\left(M-X_{Gly}\right)-exp\left(k_{20}-\frac{E_{a2}}{R}\frac{1}{T}\right)C_{cat}X_{gly}^{2}$$

The resulting fits for both catalysts are displayed in Figure 18. Table 8 includes all relevant parameters for both second-order kinetic constants, direct -1- and inverse -2-, together with key goodness-of-fit statistical parameters. The QH second-order kinetic model fits all data at the tested temperatures in an almost perfect way, with two temperature intervals for the impregnated catalyst: in this case, a relatively low activation energy—as expected considering results in Figure 16B—is retrieved: 32.47 kJ·mol^−1^ for the direct reaction constant and 30.33 kJ·mol^−1^ for the inverse reaction constant. These values are within the interval expected for esterification, transesterification, and transcarbonation reactions in homogeneous catalytic systems. For the higher temperature interval, however, activation energies are much higher (145.16 and 196.15 kJ·mol^−1^, respectively). Although these values seem surprisingly high, in our experience, some transcarbonation systems are highly affected by temperature in relatively narrow intervals. For example, in the work by Esteban et al. [32], glycerol transcarbonation with dimethyl carbonate (DMC) experienced an enormous boost in the interval 62–70 °C—considering the turnover frequency or TOF—using potassium carbonate as a catalyst. In that paper, the activation energies for all the models tested varied between 152 and 184 kJ·mol^−1^. For the most active smectite-based catalyst, the kinetic model is, at low temperatures:(2)$$\frac{dX_{Gly}}{dt}=exp\left(1.59-3097\frac{1}{T}\right)C_{cat}\left(1-X_{Gly}\right)\left(M-X_{Gly}\right)-exp\left(0.48-3650\frac{1}{T}\right)C_{cat}X_{gly}^{2}$$While, at higher temperatures—till 110 °C, it is:(3)$$\frac{dX_{Gly}}{dt}=exp\left(38.81-17478\frac{1}{T}\right)C_{cat}\left(1-X_{Gly}\right)\left(M-X_{Gly}\right)-exp\left(54.78-23605\frac{1}{T}\right)C_{cat}X_{gly}^{2}$$The kinetic model for the raw clay is established as:(4)$$\frac{dX_{Gly}}{dt}=exp\left(5.40-5250\frac{1}{T}\right)C_{cat}\left(1-X_{Gly}\right)\left(M-X_{Gly}\right)-exp\left(8.19-6468\frac{1}{T}\right)C_{cat}X_{gly}^{2}$$

In this case, the activation energies are 43.63 and 53.75 kJ·mol^−1^. Therefore, the thermal activation phenomenon is dramatically affecting the active phase activity that is created during K_2_CO_3_ calcination (K_2_O) in the high-temperature interval.

## 3. Materials and Methods

### 3.1. Materials

In the course of the experimental work, a range of reagents was employed, primarily purchased from Sigma-Aldrich, unless otherwise specified. These included extra-pure glycerol (with a 99.88% assay grade) from Fischer Chemical and propylene carbonate (with a purity of 99%), generously provided by UBE Corporation Europe (Madrid, Spain), in its synthetic-grade form. The chemicals used for the basic catalyst were potassium carbonate (K_2_CO_3_) and raw Tunisian clay. The clay was impregnated and subsequently calcined for catalyst preparation. To calibrate the HPLC analysis, the following materials were utilized: glycerol carbonate (with a purity of ≥99.5%) from Sigma-Aldrich (St. Louis, MO, USA), HPLC-grade methanol (with a test grade and a purity of 99.99%) from Scharlau Chemie SL (Barcelona, Spain), and anhydrous ethylene glycol (with a purity of 99.8%) from Sigma-Aldrich (St. Louis, MO, USA). Additionally, citric acid ACS reagent (with a purity of ≥99.5%) from Sigma-Aldrich (St. Louis, MO, USA) was employed as an internal standard for ion exclusion HPLC.

### 3.2. Methods

#### 3.2.1. Purification of Clay

The clay used in this investigation was procured from Mountain Hidoudi of the central region of Gabès, located in southeastern Tunisia [33,34,35]. The purification procedure for the raw clay involved a series of distinct unit operations: (1) raw clay milling in an agate mortar to reduce the particle size to no more than 1 mm; (2) separation of the fine fraction, with particles of less than 2 μm, by sieving; (3) Dispersion of the natural raw clay (1 kg) in distilled water to eliminate any residual remnants of animal fossils and plant debris; and (4) drying at 100 °C in an oven for 2 days. If necessary, the clay was sieved to obtain a particle size fraction with a diameter less than 100 μm. This process ensured the removal of any impurities or larger particles.

#### 3.2.2. Catalyst Preparation

A set of four smectite-based catalysts was meticulously prepared through the application of the incipient wetness impregnation method, with each catalyst possessing varying loadings of K_2_CO_3_ (ranging from 10% to 40% by weight) [36,37,38]. This method commenced with the suspension of 3 g of raw clay into 100 mL of deionized water, suspending this mixture by stirring at 400 r.p.m. Subsequently, a K_2_CO_3_ solution was thoughtfully prepared and slowly introduced into the solid suspension, leading to an intricately stirred mixture that was maintained at 343 K for a duration of 5 h. As the reaction reached its conclusion, the precipitate was filtered and then left to air dry overnight at 293 K. The resulting powder was subjected to a meticulous calcination process at 823 K in air for 3h. To delve further into the effects of the key variables, one subset of the material, a priori ideal due to its high K_2_CO_3_ loading (40%), underwent calcination at four distinct temperatures: 723, 823, 923, and 1023 K.

#### 3.2.3. Catalyst Characterization

Powder X-ray diffraction (XRD) patterns for the various samples were obtained using a Rigaku/D/MAX 2200 diffractometer, with CuK radiation generated by a Cu X-ray tube operating at 40 kV/40 mA at room temperature. All measurements were conducted in the Bragg angle (°) range of 5–75 degrees.

For the analysis of the main functional groups, FTIR spectra of the catalysts were recorded in the wavenumber region of 4000–500 cm^−1^ using a Perkin-Elmer Spectrum 100 instrument (Waltham, MA, USA).

SEM micrographs of the K_2_CO_3_-impregnated smectite particles were captured with a JEOL (Akishima, Tokio, Japan) JSM 6400 scanning electron microscope. The elemental chemical composition of the samples was determined at room temperature using the energy-dispersive X-ray spectroscopy (EDS) detector of the JEOL SEM.

The specific surface area of the synthesized catalysts was calculated using the BET method with a BELSORP MINI II instrument from the BEL Company in Japan (now part of Verder Scientific GmbH, Haan, Germany). Before conducting surface area measurements, the catalysts were degassed in an oven at 393 K for 24 h.

To characterize the basicity and basic strength of the catalysts, a similar approach was employed for Brönsted basicity tests. In this experimental setup, 100 mg of the sample was placed in 10 mL of an aqueous solution containing KCl at a concentration of 10 g/L. The pH was measured using a pH meter and a Crison 22 GLP ionometer until a stable reading was achieved (first measurement). This was followed by sonication to ensure a more intimate solid-liquid contact, promoting the mass transfer and ion exchange of cations and anions on the clay’s inner surface (second measurement). The Brönsted basicity was quantified in millimoles of hydroxide-like species per dry gram of catalyst, taking into account the hydroxides released as available strong bases.

### 3.3. The Transesterification of Glycerol-to-Glycerol Carbonate

The transesterification process of glycerol with propylene carbonate was executed within a carefully controlled glass reactor, ensuring a homogeneous reaction phase at temperatures exceeding 70 °C. The experimental protocol commenced with the introduction of 12 mL of glycerol into the reactor, gradually heating it to the desired reaction temperature while maintaining continuous stirring at 600 rpm. The reaction initiation involved the addition of 40 mL of propylene carbonate and the catalyst into the reactor.

To identify the most suitable basic catalyst, a comprehensive exploration of K_2_CO_3_’s impact on smectite and variations in the catalyst’s calcination temperature was undertaken. Once the optimal catalyst was determined, a detailed investigation was conducted under various reaction conditions. This included systematic adjustments to the reaction temperature across a range of values (344, 354, 364, 374, 384, and 394 K), fine-tuning the molar ratio of propylene carbonate to glycerol (ranging from 1.5:1 to 3:1), and evaluating different weight ratios of catalyst to glycerol (spanning from 2% to 6 wt%). All experiments were conducted under atmospheric pressure conditions, and the reaction duration was consistently set at 6 h.

It is noteworthy that the observed yields remained consistent across varying stirring speeds within the range of 400–1000 rpm, thereby ensuring that, if the stirring rate remained at a constant 600 rpm throughout the experimental process, no mass transfer limitations would be present. This study design aimed to unveil the optimal catalyst and reaction conditions for the transesterification of glycerol with propylene carbonate (Figure 19).

### 3.4. Catalyst Reusability

To assess the reusability of the prepared catalyst, the transesterification process was replicated under optimal conditions consecutively, for five cycles in total. Following each cycle, the catalyst was separated from the liquid medium using centrifugation for 5 min at 10,000× *g*. Subsequently, the retrieved catalyst underwent a thorough cleaning process involving methanol (two consecutive contacting steps with 10 mL of methanol per gram of dry smectite followed by centrifugation for 5 min at 10,000× *g*)—to remove any unreacted glycerol and propylene carbonate as well as product within the pores—and pure acetone washing (operation conditions identical to the those with methanol)—to remove methanol, which could react, together with remaining traces of reagents and products and to facilitate the solid drying. Once cleansed, the catalyst was dried at 314 K for 12 h, rendering it ready for reuse in the subsequent reaction cycle. This rigorous testing aimed to confirm the catalyst’s durability and sustainable performance over multiple cycles.

### 3.5. Glycerol Carbonate Analysis

#### 3.5.1. Qualitative Analysis

A comprehensive qualitative analysis of the compounds within the reaction liquid was conducted employing nuclear magnetic resonance (NMR) technology using a BRUKER Ultra Shield Plus 400 MHz instrument (Billerica, MA, USA). For the ^1^H NMR analysis, a sample was meticulously prepared by mixing 50 µL of the liquid sample with 1000 µL of deuterated DMSO-d4 in a 1.5 mL microtube. Subsequently, the sample underwent centrifugation at 14.000 r.p.m. (equivalent to 20,913× *g* with the rotor) and was then filtered through a 0.22 μm 13 mm disc. Finally, 750 µL of the resulting filtrate was carefully transferred into an NMR tube, which was subsequently submitted to a centralized NMR facility for analysis. This precise methodology allowed for a detailed examination of the compounds present in the reaction medium.

#### 3.5.2. Quantitative Analysis

In the ion exclusion HPLC method utilized, an initial 250 µL sample was first centrifuged at 14,000 r.p.m. to effectively remove most solid particles. Subsequently, 100 µL of the centrifuged sample was extracted and diluted 25 times using an aqueous solution containing 8 g/L of citric acid, which served as an internal standard. The resulting diluted sample then underwent filtration through a 0.22 μm 13 mm disc before being subjected to analysis by ion exclusion HPLC using a JASCO 2000 instrument equipped with a refractive index detector. Milli-Q acidic water (0.005 N H_2_SO_4_) was employed as the mobile phase, maintaining a constant flow rate of 0.5 mL/min. The separation of the peaks corresponding to each component was achieved through the use of a Rezex ROA-Organic Acid H+ (8%) column (150 × 7.80 mm). Subsequent to estimating the peak area and applying linear internal standard curves for glycerol at different reagent molar ratios, the analysis was completed. This method facilitated the accurate quantification of various components within the samples.

Following the estimation of raw glycerol conversions in all experimental runs, we employed Origin 2021 software to enhance the data’s accuracy by implementing interpolation with hyperbolic functions. This smoothing process effectively minimized the impact of experimental random errors and resulted in refined values for glycerol conversion. These corrected data sets were then utilized in the subsequent kinetic modeling analysis, allowing for a more precise assessment of the reaction kinetics.

#### 3.5.3. Statistical Methods

Kinetic data retrieved at diverse catalyst concentrations, reagent molar ratios, and temperature values are employed to propose, fit, and validate a potential kinetic model based on the assumption of an elemental bimolecular reaction between glycerol and ethylene carbonate. Fitting of the kinetic model to diverse datasets is performed using a combination of numerical integration of the ordinary differential equation (ODE) of the model with a variable step Euler method and a nonlinear regression algorithm named NL2SOL, an evolved form of the Levenberg-Marquardt gradient method. These algorithms are implemented in the Aspen Custom Modeler version 12.1 software [32].

For the final comparison between the most active K_2_CO_3_ smectite-based catalyst and the original, raw smectite, non-isothermal kinetic modeling was performed. Considering the behavior of the impregnated smectite, the 70–110 °C interval was studied, while for the raw smectite, a higher temperature interval was set: 100–120 °C. In both cases, a 2% w/w catalyst concentration and a propylene carbonate-to-glycerol molar ratio (M) of 2 was chosen. Assuming that the process is reversible, with direct and inverse reactions, the kinetic constants were expressed in terms of their Arrhenius equations (Equation (5)) to facilitate the nonlinear regression calculations and obtained the neperian logarithms of the preexponential factor and the activation energies for the kinetic constants of both direct and inverse reactions.
(5)$$k=exp\left(lnk_{0}-\frac{E_{a}}{R}\frac{1}{T}\right)$$

Apart from physical criteria (positive values for the kinetic constants and a positive value for the activation energy of all kinetic constants within the 10–300 kJ·mol^−1^ interval), several statistical criteria were analyzed to determine if the model fitting is correct, either in with fresh or with reused catalysts. Statistical analysis was based on the standard error for the kinetic parameters of each model and the goodness-of-fit parameters based on the least-squares method. In fact, all goodness-of-fit statistical criteria applied are based on minimizing the sum of variances or Residual standard error (RSE), as indicated in Equation (6), where X_e_ stands for the experimental values of glycerol conversions, while X_c_ is the conversion value calculated with the kinetic model. A derivate parameter is the standard error of estimate (SEE), a parameter that is calculated by using Equation (7). Fisher’s F value (F) is estimated through Equation (8) and should be higher than a threshold value for the given number of data and kinetic parameters (for our case, the critical values for F are in the range 20–30). A fine fitting means a low value of RSS and, in consequence, for RSE, while the F-value should be high and, in any case, higher than the aforementioned threshold values at 95% confidence.
(6)$$\mathrm{RSE}=\overset{N}{\underset{i=1}{\sum }}{\left(X_{e}-X_{c}\right)}_{i}^{2}$$
(7)$$\mathrm{SSE}=\frac{1}{N}\sqrt{\overset{N}{\underset{i=1}{\sum }}{\left(X_{e}-X_{c}\right)}_{i}^{2}}$$
(8)$$F-\mathrm{value}=\frac{\sum_{n=1}^{N}\frac{{\left(X_{c}\right)}^{2}}{K}}{\sum_{n=1}^{N}\frac{{\left(X_{e}-X_{c}\right)}^{2}}{N-K}}$$

Finally, to analyze the performance of the model in terms of dX/dt, the percentage of explained variable (VE) is the most adequate goodnessoffit. This parameter indicates the capacity of the model to calculate a minimal conversion change (dX) with a small time variation (dt), and it can be estimated with Equation (9).
(9)$$\mathrm{VE}\;\left(\%\right)=100\cdot \left[1-\frac{\sum_{n=1}^{N}{\mathrm{SSQ}}_{i}}{\sum_{n=1}^{N}{\mathrm{SSQ}}_{\mathrm{mean}\;i}}\right]$$

## 4. Conclusions

The synthesis of glycerol carbonate via the transcarbonation of glycerol with propylene carbonate with raw Tunisian smectite and K_2_CO_3_-impregnated Tunisian smectite as catalysts was studied. Diverse DRX, SEM-EDS, and FTIR analyses allowed for the mineralogical and microstructural characterization of the raw and calcined clays, while further porosimetry and acid-based titration allowed for an understanding of their inner structure and potential capacity as basic catalysts. Catalytic tests with the reaction under study permitted the selection of a smectite-based catalyst impregnated with 40% K_2_CO_3_ and calcined at 550 °C, as it showed the highest activity. Further quantitative tests allowed for the understanding of the effect of catalyst concentration, reagent molar ratio, and temperature on the transcarbonation reaction under study for both the raw and the most active calcined basic smectite catalysts, with a final in-depth comparison of their performance based on the fitting of a quasi-homogeneous reversible second-order model to data retrieved for both catalysts working at diverse temperatures. In addition, both catalysts showed notable stability for four reaction cycles, with the raw smectite less active but slightly more stable.

## Figures and Tables

**Figure 1 ijms-25-02442-f001:**
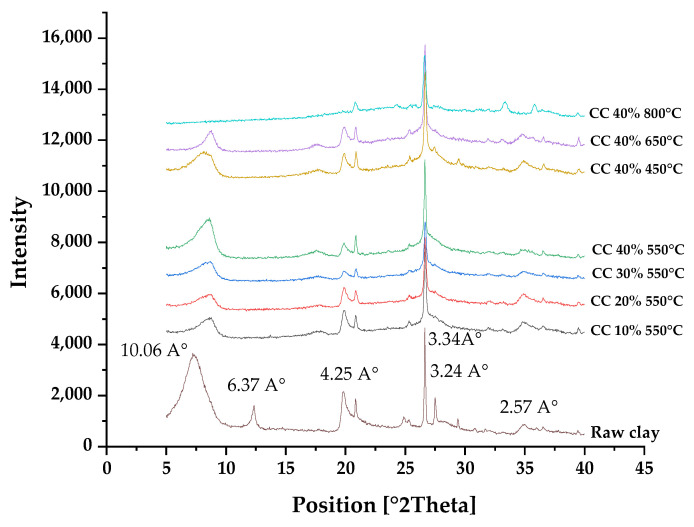
XRD patterns of smectite. Smectite catalysts were impregnated with different levels of potassium carbonate loading (10 to 40% by weight) and calcination temperature (450–800 °C) during calcination and prior to the transesterification reaction.

**Figure 2 ijms-25-02442-f002:**
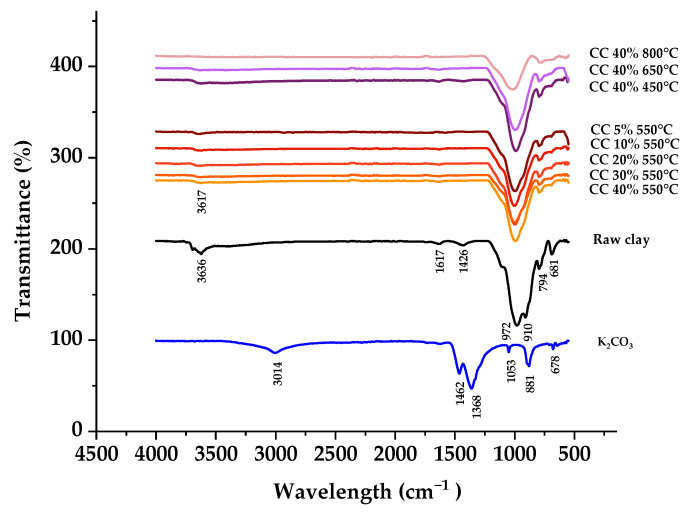
FTIR spectra of smectite, K_2_CO_3_, and smectite-supported catalysts impregnated with different potassium carbonate loading levels (10–40 wt%) and calcination temperature (450–800 °C) during calcination and before the transesterification reaction.

**Figure 3 ijms-25-02442-f003:**
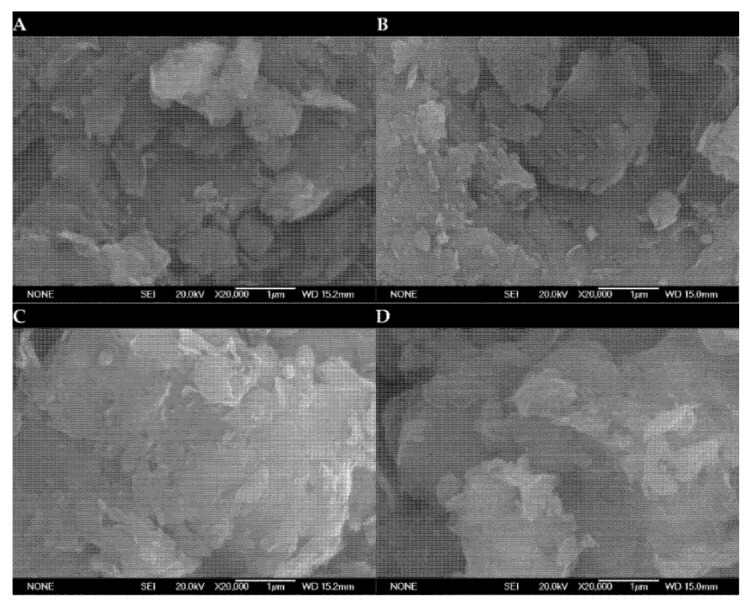
SEM micrographs of (**A**) K_2_CO_3_ (10%)/smectite; (**B**) K_2_CO_3_ (20%)/smectite; (**C**) K_2_CO_3_ (30%)/smectite; (**D**) K_2_CO_3_ (40%)/smectite catalysts with a calcination temperature of 550 °C at 20,000× magnification.

**Figure 4 ijms-25-02442-f004:**
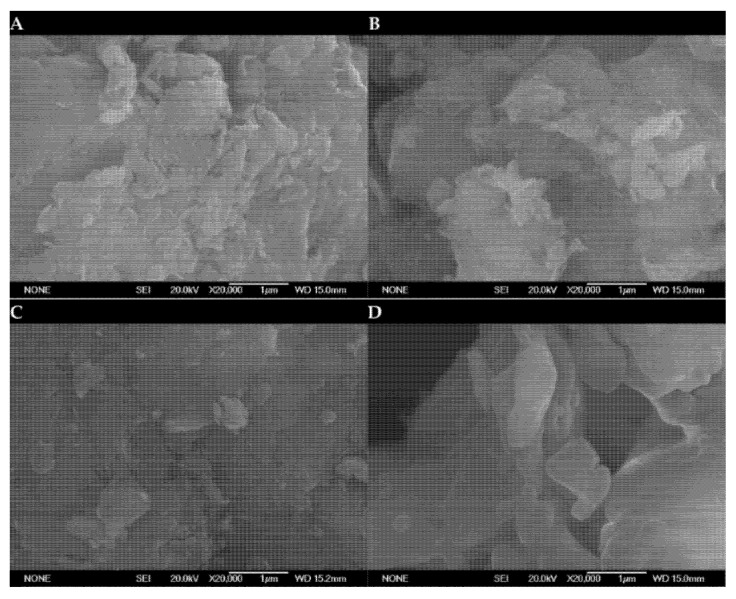
SEM micrographs of (**A**) CC at 450 °C; (**B**) CC at 550 °C; (**C**) CC at 650 °C; (**D**) CC at 650 °C catalysts with 40% by weight of K_2_CO_3_ at 20,000× magnification.

**Figure 5 ijms-25-02442-f005:**
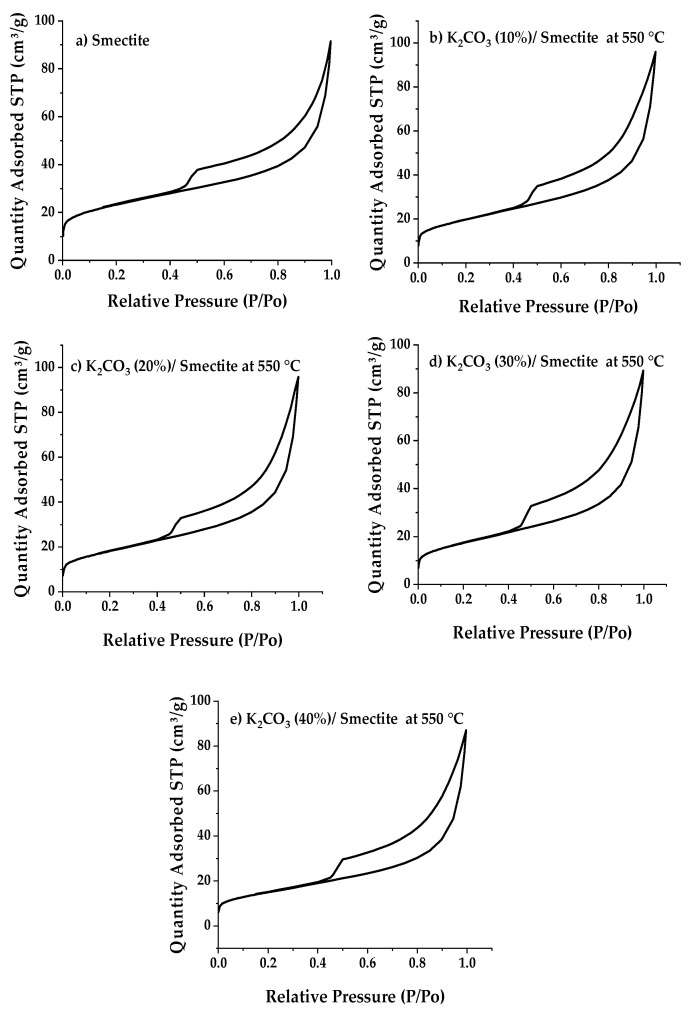
Nitrogen adsorption-desorption isotherms of smectite catalysts impregnated with different potassium carbonate loading levels: (**b**) 10%*w*/*w* dry solid, (**c**) 20%*w*/*w* dry solid, (**d**) 10%*w*/*w* dry solid, and (**e**) 10%*w*/*w* dry solid to 40%*w*/*w* calcined at 550 °C in comparison to the raw or native clay (**a**).

**Figure 6 ijms-25-02442-f006:**
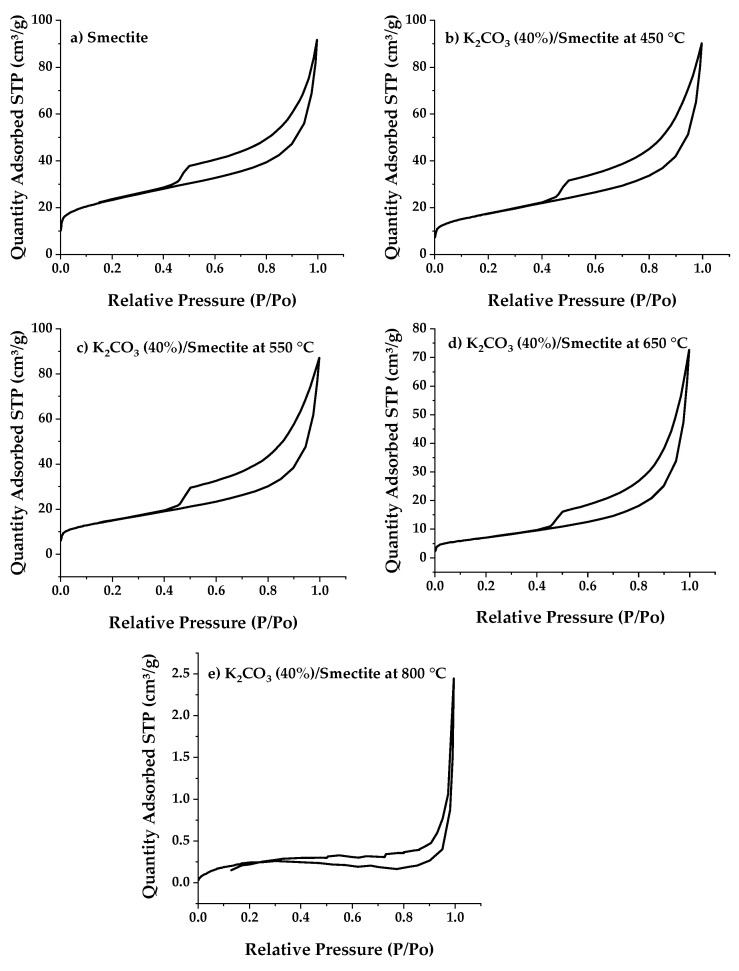
Nitrogen adsorption-desorption isotherms of impregnated smectite catalysts at different calcination temperatures: (**b**) 450 °C, (**c**) 550 °C, (**d**) 650 °C, and (**e**) 800 °C, in comparison to the raw clay (**a**).

**Figure 7 ijms-25-02442-f007:**
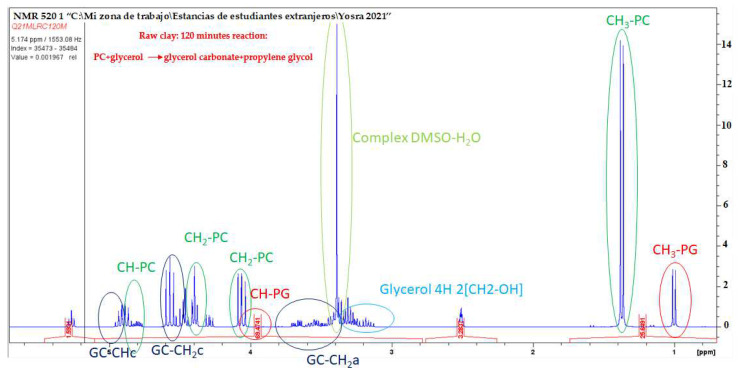
Expanded region (3.2 to 5.3 ppm) of ^1^H NMR spectra in DMSO of a reaction sample at 120 min containing the mixture of Gly, EC, GC, and EG. The catalyst here used was raw clay (RC).

**Figure 8 ijms-25-02442-f008:**
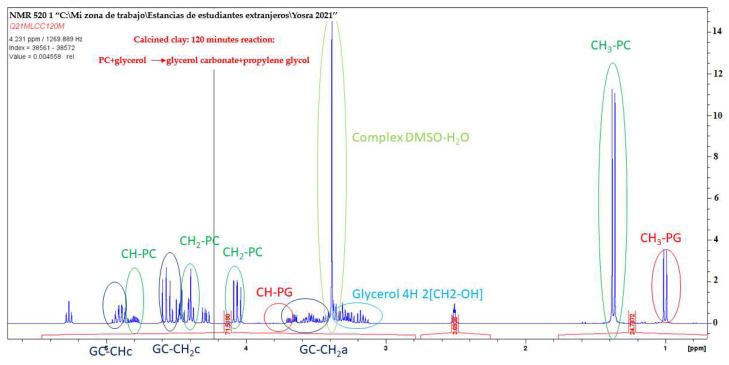
Expanded region (3.2 to 5.3 ppm) of ^1^H NMR spectra in DMSO of a reaction sample catalyzed by the most active K_2_CO_3_-modified smectite (CC) containing the mixture of Gly, EC, GC, and EG.

**Figure 9 ijms-25-02442-f009:**
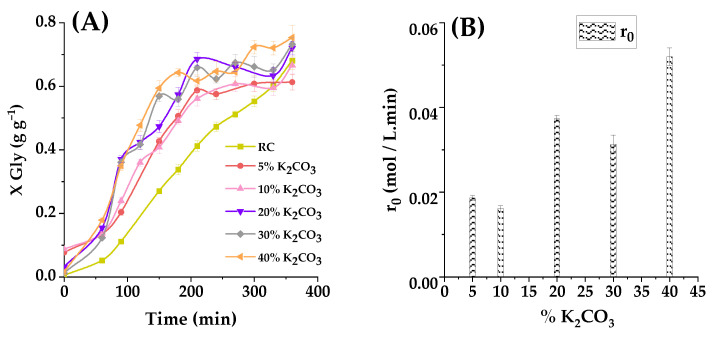
(**A**) Effect of K_2_CO_3_ loading impregnated on smectite at 100 °C with a molar ratio of PC/Gly 2:1. The conversion of glycerol is presented as a function of reaction time with 2% in quantity catalyst (RC or CC).Where ■ represents the RC (0% in charge of K_2_CO_3_), ● loading of 5% potassium carbonate, ▲ a K_2_CO_3_ loading of 10%, ▼ 20% loading, ◆ 30% loading, and ◀ 40% K_2_CO_3_ loading.(**B**) Initial reaction rates observed for these runs.

**Figure 10 ijms-25-02442-f010:**
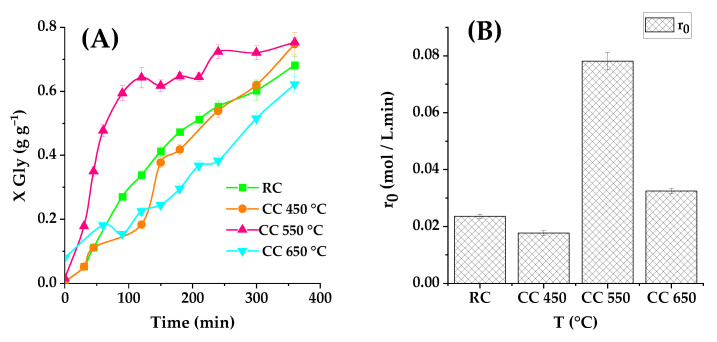
(**A**) Effect of calcination temperature with a K_2_CO_3_ load equal to 40% at 100 °C, with a molar ratio of PC/Gly 2:1. The conversion of glycerol is presented as a function of the reaction time with 2% in the quantity of catalyst (RC or CC), where ■ represents the raw clay; ● a calcination temperature equal to 450 °C; ▲ a calcination temperature equal to 550 °C; and ▼ represents a temperature of 650 °C. (**B**) Initial reaction rates observed for these experiments.

**Figure 11 ijms-25-02442-f011:**
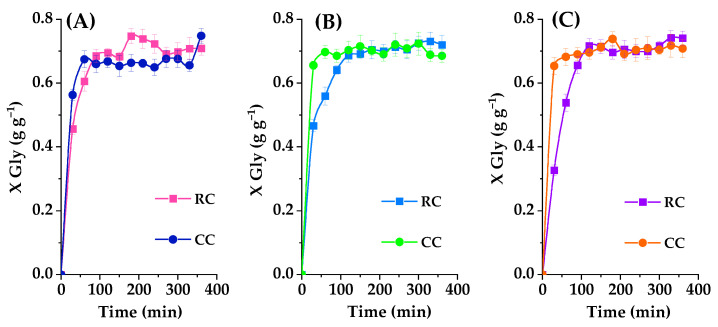
Effect of the quantity of catalyst, taking into account the effect of the nature of the catalyst (RC or CC) at 120 °C with a molar ratio of 2:1. The conversion of glycerol is presented as a function of the time of reaction with different catalyst loadings, where (**A**) represents a catalyst loading of 2%, (**B**) represents a catalyst loading of 4%, and (**C**) represents a catalyst loading of 6%.

**Figure 12 ijms-25-02442-f012:**
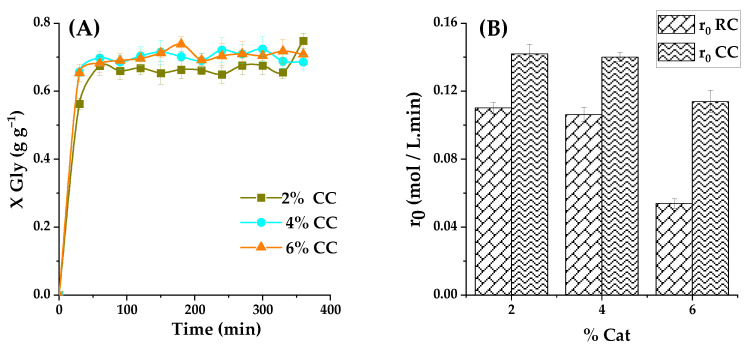
(**A**) Effect of CC loading at 120 °C and a CE/Gly ratio of 2 on Gly conversion as a function of reaction time with different catalyst loadings, where ■ represents a loading of 2% catalyst, ● a catalyst loading of 4%, and ▲ a catalyst loading of 6%. (**B**) Initial reaction rates recorded for these experiments.

**Figure 13 ijms-25-02442-f013:**
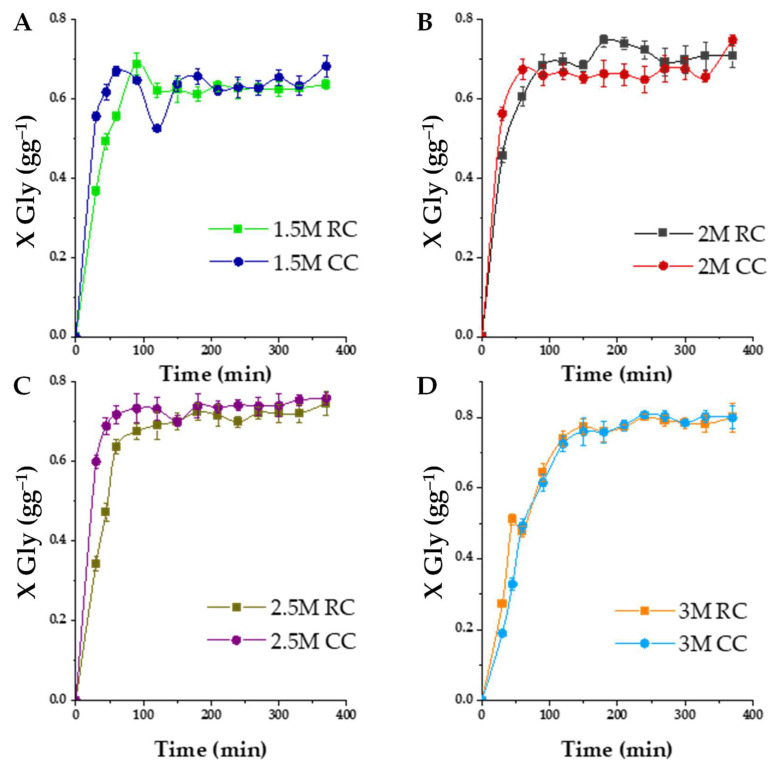
Effect of the initial molar ratio of the reactants, taking into account the effect of the nature of the catalyst (RC or CC) at 120 °C with a load of 2%. The conversion of glycerol is presented as a function of the reaction time with the PC/Gly ratio (M), where (**A**) M = 1.5, (**B**) M = 2, (**C**) M = 2.5, and (**D**) M = 3.

**Figure 14 ijms-25-02442-f014:**
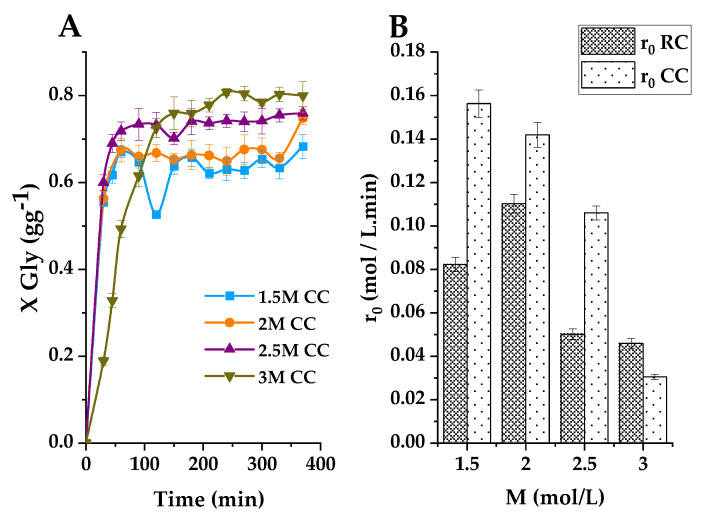
(**A**) Effect of initial molar ratio of reactants using CC at 120 °C, with 2% loading on glycerol conversion as a function of reaction time with CE/Gly ratio (M), where ■ M = 1.5, ● M = 2, ▲ M = 2.5, and ▼ M = 3. (**B**) Effect of initial reaction speed with initial PC/Gly molar.

**Figure 15 ijms-25-02442-f015:**
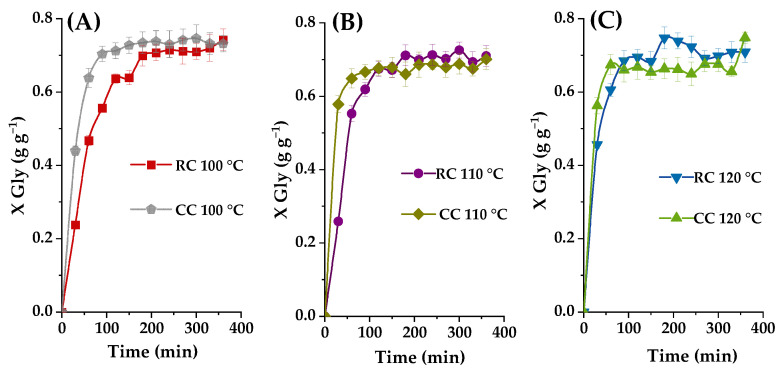
Effect of the initial molar ratio of the reagents, taking into account the effect of the nature of the catalyst (RC or CC), with 2% load and with 2M. The conversion of glycerol is presented as a function of the reaction time with the ratio PC/Gly (M), where (**A**) T = 100, (**B**) T = 110, and (**C**) T = 120 °C.

**Figure 16 ijms-25-02442-f016:**
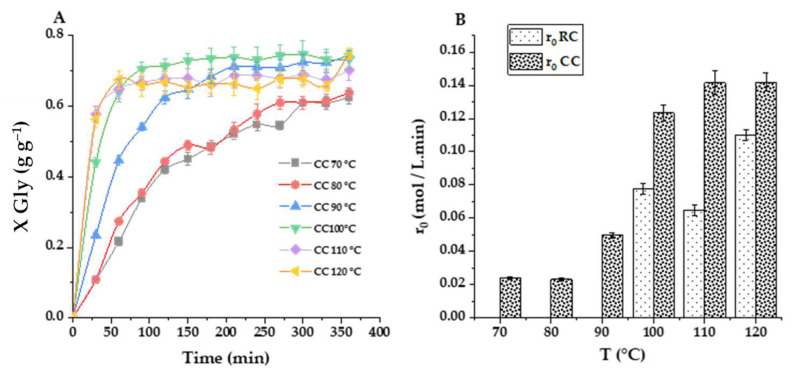
(**A**) Effect of initial molar ratio of reactants using CC at 120 °C, with 2% loading on glycerol conversion as a function of reaction time with CE/Gly ratio (M), where ■ T = 70 °C, ● T = 80 °C, ▲ T = 90 °C, ▼ T = 100 °C, ◆ T = 110 °C, and ◀ T = 120 °C. (**B**) Effect of the initial reaction speed with the initial PC/Gly molar.

**Figure 17 ijms-25-02442-f017:**
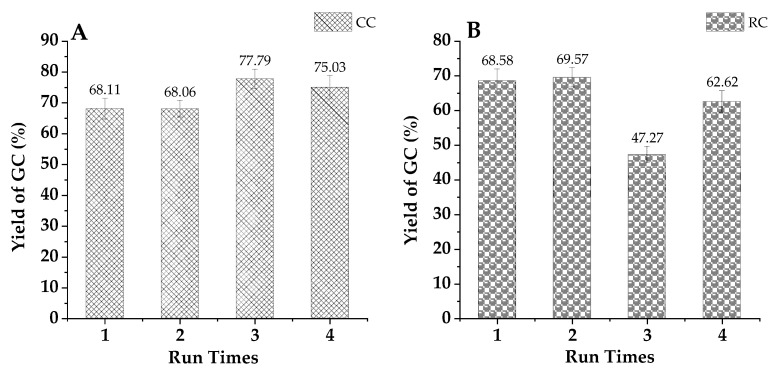
Reuse cycles of raw (**A**) and calcined (**B**) smectite catalyst (40% in K_2_CO_3_ at 550 °C) at 120 °C (standard error percentage range fluctuates between ±0.5 and 5%).

**Figure 18 ijms-25-02442-f018:**
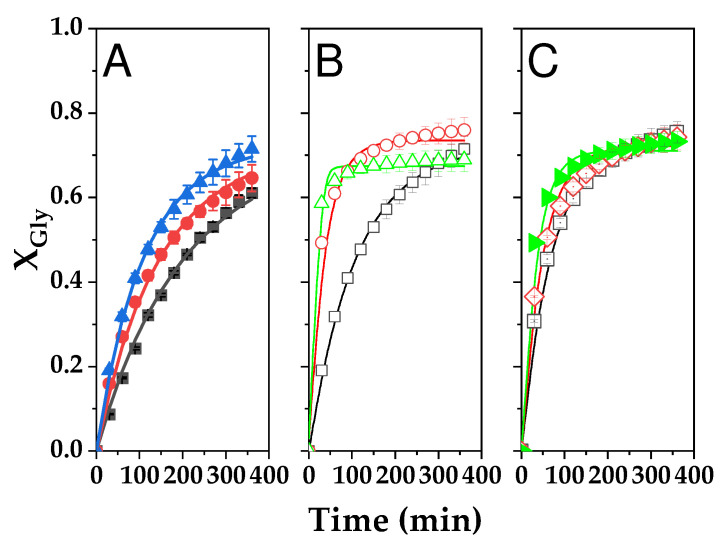
Fitting of the quasi-homogeneous kinetic model to glycerol conversion data (X_Gly_) in the transesterification reaction of glycerol with propylene carbonate driven by K_2_CO_3_-impregnated Tunisian smectite (CC) and raw clay (RC). Dots represent experimental data, while lines reflectthe fit of the model for each experimental data point. (**A**) Effect of the temperature on CC catalyst activity in the low-temperature interval (■ 70 °C; ● 80 °C; ▲ 90 °C); (**B**) Temperature influence on CC catalyst activity in the high-temperature range (☐ 90 °C; ❍ 100 °C; △ 110 °C); and (**C**) Temperature effects on raw smectite (RC) activity in the high-temperature range (▣ 100 °C; ◈ 110 °C; ▶ 120 °C). Common conditions: 2% *w*/*w* catalyst load and M = 2.

**Figure 19 ijms-25-02442-f019:**
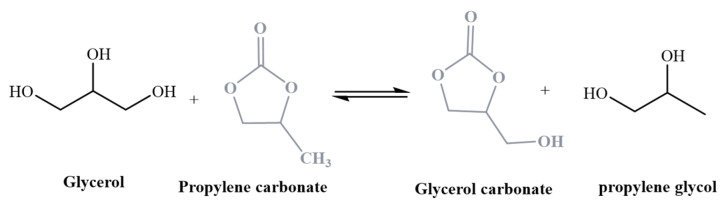
Transcarbonation reaction catalyzed by smectite modified with K_2_CO_3_ of glycerol and propylene carbonate to obtain monopropylene glycol and glycerol carbonate.

**Table 1 ijms-25-02442-t001:** XRD model data for smectite catalysts impregnated with different levels of potassium carbonate loading (10 to 40% by weight) and calcined at 550 °C.

Pos. [°2Th.] RC	Spacing d [Å]	Pos. [°2Th.] 10%	Spacing d [Å]	Pos. [°2Th.] 20%	Spacing d [Å]	Pos. [°2Th.] 30%	Spacing d [Å]	Pos. [°2Th.] 40%	Spacing d [Å]
8.78	10.06	8.82	10.01	5.1	17.13	8.77	10.07	7.70	11.47
13.89	6.37	13.70	6.46	8.81	10.03	17.63	5.02	8.57	10.31
15.81	5.60	14.88	5.95	17.79	4.98	19.82	4.47	17.50	5.06
17.66	5.02	17.77	4.99	19.85	4.47	20.85	4.25	19.85	4.47
19.82	4.47	19.81	4.48	20.85	4.25	25.33	3.51	20.85	4.26
20.87	4.25	20.86	4.25	25.36	3.51	26.64	3.34	21.41	4.15
23.50	3.78	23.54	3.77	26.64	3.34	31.98	2.79	22.06	4.02
25.36	3.51	25.33	3.51	27.47	3.24	33.15	2.70	23.62	3.76
26.66	3.34	26.64	3.34	31.97	2.79	34.84	2.57	25.30	3.51
27.52	3.24	27.56	3.23	33.10	2.70	36.53	2.45	26.63	3.34
30.83	2.89	31.95	2.80	34.84	2.57	39.4	2.28	32.05	2.79
33.17	2.70	33.15	2.70	36.53	2.45			33.10	2.70
34.81	2.57	34.76	2.58	39.44	2.28			34.65	2.58
36.57	2.45	36.54	2.45					35.31	2.54
37.88	2.37	39.46	2.28					36.53	2.45
39.48	2.28							37.79	2.38

**Table 2 ijms-25-02442-t002:** XRD model data for smectite catalysts impregnated with 40% K_2_CO_3_ at different calcination temperatures (450–800 °C).

Pos. [°2Th.] RC	Spacing d [Å]	Pos. [°2Th.] 450 °C	Spacing d [Å]	Pos. [°2Th.] 650 °C	Spacing d [Å]	Pos. [°2Th.] 800 °C	Spacing d [Å]
8.78	10.06	6.33	13.94	8.75	10.09	18.25	4.85
13.89	6.37	8.09	10.92	17.58	5.04	19.80	4.48
15.81	5.60	8.75	10.10	19.78	4.48	20.78	4.27
17.66	5.02	17.28	5.13	20.84	4.26	24.24	3.67
19.82	4.47	17.76	4.99	23.53	3.78	25.38	3.50
20.87	4.25	19.88	4.46	25.34	3.51	25.78	3.45
23.50	3.78	20.87	4.25	26.6542	3.34	26.60	3.35
25.36	3.51	23.53	3.77	31.92	2.80	27.37	3.25
26.66	3.34	25.37	3.51	33.12	2.70	27.54	3.23
27.52	3.24	26.66	3.34	34.85	2.57	29.93	2.98
30.83	2.89	27.44	3.25	35.69	2.51	31.08	2.87
33.17	2.70	29.45	3.03	36.57	2.45	31.94	2.80
34.81	2.57	32.04	2.79	39.50	2.28	33.32	2.68
36.57	2.45	33.29	2.69			35.84	2.50
37.88	2.37	34.80	2.57			39.42	2.28
39.48	2.28	36.58	2.45				

**Table 3 ijms-25-02442-t003:** EDS data of smectite catalysts impregnated with different potassium carbonate loading levels (10 to 40% by weight) at calcination temperature (450 °C–800 °C).

Spectrum	C	O	Na	Mg	Al	Si	K	Ca	Fe
Raw clay	34.71	51.92	0.67	0.50	3.51	7.33	0.31	0.35	0.71
Spect 10%-550 °C	46.43	41.55	0.00	0.28	5.11	5.56	0.53	0.00	0.55
Spect20%-550 °C	25.80	52.55	0.26	0.57	4.61	11.14	1.50	0.00	3.56
Spect 30%-550 °C	41.50	45.52	0.53	0.42	3.09	7.26	0.83	0.13	0.73
Spect40%-550 °C	39.35	46.91	0.00	0.53	3.35	7.75	0.84	0.52	0.74
Spect 40%-450 °C	42.00	46.46	0.00	0.44	2.88	6.23	0.80	0.58	0.60
Spect 40%-650 °C	51.59	40.37	0.00	0.28	2.21	4.30	0.55	0.11	0.59
Spect 40%-800 °C	34.71	51.92	0.67	0.50	3.51	7.33	0.31	0.35	0.71

**Table 4 ijms-25-02442-t004:** Porosimetry characteristics of smectite catalysts impregnated with different potassium carbonate loading levels (10 to 40% by weight) at 550 °C.

		Raw Clay	K_2_CO_3_ (10%)/Smectite	K_2_CO_3_ (20%)/Smectite	K_2_CO_3_ (30%)/Smectite	K_2_CO_3_ (40%)/Smectite
**Area (m^2^;/g)**	BET area:	74.235	61.5113	55.8759	52.9210	45.6611
**Pore volume (cm^3^;/g)**	Total pore volume of adsorption at a pore point 6925.979 Å diameter at P/P_0_ = 0.997217394 cm^3^;/g	0.141	0.1484	0.14831	0.1380	0.1345
**Pore size (** **Å)**	Average adsorption pore width (4 V/A per BET)	76.3032	96.5361	106.177	104.3311	117.890

**Table 5 ijms-25-02442-t005:** Porosimetry characteristics of smectite catalysts impregnated with 40% potassium carbonate at different temperatures (450–800 °C) during calcination and before the transesterification reaction.

		Raw Clay	K_2_CO_3_ (40%)/Smectite at 450 °C	K_2_CO_3_ (40%)/Smectite at 550 °C	K_2_CO_3_ (40%)/Smectite at 650 °C	K_2_CO_3_ (40%)/Smectite at 800 °C
**Area (m^2^;/g)**	BET area:	74.4	53.83	45.66	21.32	0.491
**Pore volume (cm^3^;/g)**	Total pore volume of adsorption at a pore point 6925.979 Å diameter at P/P_0_ = 0.997217394 cm^3^;/g	0.141	0.139	0.135	0.112	0.0037
**Pore size (** **Å)**	Average adsorption pore width (4 V/A per BET)	76.3	103.67	117.9	210.7	308.3

**Table 6 ijms-25-02442-t006:** Value of the pH of catalysts obtained by K_2_CO_3_-smectite contact before and after ultrasound pretreatments when the solids are suspended in Milli-Q water.

Name of the Catalyst	pH before US Treatment	pH after US Treatment
Raw clay	8.84	8.58
10% at 550 °C	9.65	8.76
20% at 550 °C	9.82	8.95
30% at 550 °C	9.99	9.2
40% at 550 °C	9.92	9.14
40% at 450 °C	9.27	8.63
40% at 650 °C	7.87	7.92
40% at 800 °C	7.35	7.36

**Table 7 ijms-25-02442-t007:** Relationships between Brönsted basicity estimated from pH values after US treatment in Table 6 and expressed as mmol of hydroxide-like species per gram of dry solid, BET solid specific surface, initial reaction rate (r_0_), and turnover frequency (TOF) in terms of mol of glycerol converted per mole of OH^−^ and second.

Catalyst	C_OH_^−^ in Solid [mmol/g Solid]	BET Specific Surface [m^2^/g]	r_0_ [mol_CG_/(L·min)]	TOF [s^−1^]
RC	3.80 × 10^−4^	74.24	0.007	12.18
CC 10% at 550 °C	5.75 × 10^−4^	61.31	0.017	19.54
CC 20% at 550 °C	8.91 × 10^−4^	55.87	0.036	26.71
CC30% at 550 °C	15.85 × 10^−4^	52.92	0.034	14.19
CC 40% at 550 °C	13.80 × 10^−4^	45.66	0.051	24.44

**Table 8 ijms-25-02442-t008:** Statistical parameters calculated by fitting the quasi-homogeneous second-order model to different sets of experimental data of the best K_2_CO_3_-smectite (CC) and raw clay (RC) at varying temperatures. Residual sum of squares (RSS), standard error of estimate (S_e_), percentage of variation explained (VE%), and Fisher’s F value (F) at 95% confidence.

Catalyst/Conditions	Ln k_10_	E_a1_/R	Ln k_20_	E_a2_/R	RSS	S_e_	VE%	F
CC/70–90 °C	1.59 ± 0.56	3097 ± 204	0.48 ± 0.37	3650 ± 1344	0.0051	1.19%	99.69	15,971
CC/90–110 °C	38.81 ± 2.25	17,478 ± 822	54.78 ± 3.42	23,605 ± 1275	0.0161	2.14%	99.04	8252
RC/100–120 °C	5.40 ± 1.45	5250 ± 552	8.19 ± 2.67	6468 ± 251	0.0161	2.15%	98.98	8510

## Data Availability

All relevant data are provided in the figures and tables of this manuscript. The authors will provide any interested reader with the tables containing data and the original Origin 2021 figures.

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
