# Peer review of "K2CO3-Modified Smectites as Basic Catalysts for Glycerol Transcarbonation to Glycerol Carbonate"

_ijms, 2024, doi:10.3390/ijms25042442_

Round 1
Reviewer 1 Report
Comments and Suggestions for Authors
- Please provide some further details on the method how the NMR measurements were quantified to determine conversion rates of the reaction.
- The reaction proceeds in non-aqueous media. In contrast the the determination of the basicity was performed in aqueous media which provides the extent of -OH groups. Were there other conventional methods attempted for determination of basicity (e.g. CO2 sorption/desorption)?
- Terms conversion of glycerol and yield of gycerol carbonate are used in equivalent sense. Is it proven that formation of any side products can be fully excluded?
- The specific surface area values have been determined throughout for all the catalysts. It would be interesting to see comparison of the activity of catalysts related to unit surface area (instead of weight of catalysts on the Y axis) at least for one series.
- Even further, it might be interesting to see a comparison of activities related to unit amount of K2CO3 dispersed on the unit surface area at least for one series of catalysts (since primary role of basicity is suggested in the reaction).
- in Table 3 the heading of the last column is „Au”. Is it really related to gold (e.g. evaporated to provide good electric conductance on the surface of sample)? It is not usual that gold percentage is shown in EDS analyses.
Comments on the Quality of English LanguageThe English is fine, some typos should be corrected.
Author Response
Response to reviewer 1
We are very grateful to the reviewer for his/her nice appreciation of our review and the helpful comments and suggestions. In the following paragraphs we address each of them separately, while the text has been modified where needed using the track changes Word tool. In the following paragraphs, the reviewer can find his/her comments in black letters, our answers in blue font and the added text in the manuscript in red font.
Comment 1
“Please provide some further details on the method how the NMR measurements were quantified to determine conversion rates of the reaction.”
We would like to clarify that the conversion calculations were not derived from Nuclear Magnetic Resonance (NMR) measurements. 1H-NMR was used to identify reagents and products of the reaction and to exclude the presence of impurities in a significant level, that is, from a qualitative point of view. Instead, we utilized High-Performance Liquid Chromatography (HPLC) to generate the chromatograms required for our analysis. By measuring the glycerol concentration at various time points, we were able to determine the conversion rates of the reaction over time. We hope these clarifications meet your expectations, and we remain available for any further clarification needed. Again, HPLC suggested the absence of any impurity with a similar elution pattern of reagents and products in an appreciable concentration.
Comment 2
“The reaction proceeds in non-aqueous media. In contrast the determination of the basicity was performed in aqueous media which provides the extent of -OH groups. Were there other conventional methods attempted for determination of basicity (e.g. CO2 sorption/desorption)?”
In response to your inquiry about the determination of basicity, in addition to the aqueous analysis for evaluating -OH groups, we chose to employ a Hammett indicator method, following the detailed procedure outlined by Xie and Huang [*]. This involved simply placing approximately 300 mg of the sample into 1 ml of a Hammett indicator solution, diluted with 10 ml of methanol. After allowing a two-hour period for equilibrium without observing any additional color changes, the basic strength was defined by comparing it with standard Hammett indicators.It's worth noting that other conventional methods, such as CO2 sorption/desorption, have not been explored in the scope of this study, though we are interested to perform such analyses when we complete our research in catalysts activity and stability in continuous processes in the near future. These additional approaches could provide a more comprehensive understanding of the basicity of the synthesized catalysts. Our intention in this first study was to establish a first correlation between Brönsted basicity and catalyst activity.
[*] W. Xie, X. Huang, Catal. Lett. 107 (1–2) (2006) 53.
Comment 3
“Terms conversion of glycerol and yield of glycerol carbonate are used in equivalent sense. Is it proven that formation of any side products can be fully excluded?”
The terms glycerol conversion” and “glycerol carbonate yield” are used interchangeably in this study, reflecting the specific results of the transesterification process involving propylene carbonate and glycerol. The assurance that the formation of any side products has been completely excluded is supported by comprehensive analytical techniques, including chromatographic analyzes and NMR spectra.Chromatograms generated by HPLC meticulously detail the compounds present in the reaction mixture. The absence of peaks corresponding to undesirable by-products confirms the purity of the glycerol carbonate product. Additionally, proton NMR provide a detailed molecular fingerprint of the reaction products, allowing us to verify the absence of any unexpected chemical entities.
These rigorous analytical methods, combined with the absence of signals or peaks indicating side products in the chromatograms and NMR spectra, provide strong evidence that the transesterification process produces the desired glycerol carbonate without the formation of unwanted byproducts. Therefore, the terms “glycerol conversion” and “glycerol carbonate yield” can confidently be used synonymously to represent the successful and selective synthesis achieved in this study. »
Comment 4
“The specific surface area values have been determined throughout for all the catalysts. It would be interesting to see comparison of the activity of catalysts related to unit surface area (instead of weight of catalysts on the Y axis) at least for one series.
We are very grateful to the reviewer for his/her suggestion. There are some clarifications in the experimental section and, more in particular, at the end of the subsection “2.2.3. Catalyst characterization”. Furthermore, more comments have been added in subsection “3.1.5. Catalysts basicity” on the evolution of the basicity of the catalysts at different potassium carbonate contents and calcination temperature values. Finally, a new Table 7 compiles the values of apparent concentration of Brönsted basic sites per gram of catalyst, the BET specific surface, the initial reaction rate of the test reactions shown in Figure 9.A and the turnover frequency estimated as mol of glycerol transformed by mol of catalytic active species and second or, more simply, as [s-1], and comments on it are included in subsection “3.3.1. Effect of K2CO3 loading”.
Comment 5
“Even further, it might be interesting to see a comparison of activities related to unit amount of K2CO3 dispersed on the unit surface area at least for one series of catalysts (since primary role of basicity is suggested in the reaction).”
We appreciated very much this suggestion and have provided the indicated information in the new Table 7, adding comment is the subsection where this table is, subsection “3.3.1. Effect of K2CO3 loading”.
Comment 6
“In Table 3 the heading of the last column is „Au”. Is it really related to gold (e.g. evaporated to provide good electric conductance on the surface of sample)? It is not usual that gold percentage is shown in EDS analyses.”
Thank you for your observation. Effectively, Au is present only in minor amounts due to the need to cover the particle surface with a conductive material. We have recalculated the percentages of all relevant elements after removing gold in the new Table 3.
Comment 7
“The English is fine, some typos should be corrected.
We are grateful for the reviewer’s appreciation.
We thank the reviewer for his valuable input and constructive comments, which will contribute to the refinement and improvement of our study.

Reviewer 2 Report
Comments and Suggestions for Authors
This manuscript can be considered for publication if a revision would be carried out focusing on the issues below:
• Why was Tuisian smectite chosen for research, what are its extracted quantities?
• The name should be unified, sometimes called smectite clay, elsewhere Tunisian smectite or just clay.
• Why was the influence of the duration of transesterification of glycerol-to-glycerol carbonate not studied? A constant duration of 6 hours was chosen.
• Scheme 1 must be submitted in English.
• How much methanol and acetone were used for cleaning catalysts. Why is the catalyst dried for 12 hours? (2.4)
• Figure 1 needs to be supplemented with an intensity scale.
• Need more discussion, explanation in table 3, for example why Na is Spect 20%-550 °C and Spect 30%-550 °C, but not Spect 10%-550 °C and Spect 40%-550 °C, or Au Raw clay - 0.36, Spect 10%-550°C - 0.44 and so on...
• Pictures are repeated in Figure 5.
• Catalyst reuse - important research, and the obtained results cannot be interpreted as "suggesting a certain catalyst reactivation or a notable experimental error." If authors believe that there may be an experimental error, the study should be repeated. And if such good results are obtained, why only 4 reuse cycles were studied.
Author Response
Response to reviewer 2
We are very grateful to the reviewer for his/her nice appreciation of our review and the helpful comments and suggestions. In the following paragraphs we address each of them separately, while the text has been modified where needed using the track changes Word tool. In the following paragraphs, the reviewer can find his/her comments in black letters, our answers in blue font and the added text in the manuscript in red font.
Comment 1
“This manuscript can be considered for publication if a revision would be carried out focusing on the issues below: Why was Tuisian smectite chosen for research, what are its extracted quantities?”
Tunisian smectite was selected for our research due to its unique characteristics aligning with our study requirements. Its ample availability in significant quantities makes it a practical choice for our experiments. The properties of the clay, such as its surface area and basicity capacity, closely align with the aspects we aim to explore. Additionally, economic considerations played a role in our decision-making. If Tunisian smectite met the process needs, it represented a cost-effective option even at a larger scale.
Comment 2
“The name should be unified, sometimes called smectite clay, elsewhere Tunisian smectite or just clay.”
I appreciate the suggestion to unify the name throughout the manuscript. After careful consideration, I have decided to consistently refer to it as 'Tunisian smectite' to ensure clarity and coherence in terminology. Thank you for the guidance.
Comment 3
“Why was the influence of the duration of transesterification of glycerol-to-glycerol carbonate not studied? A constant duration of 6 hours was chosen.”
The influence of the transesterification duration of glycerol to glycerol carbonate was not investigated in this study. The decision to maintain a constant duration of 6 hours was made for several reasons. Firstly, this period was deemed sufficient to achieve reaction equilibrium while optimizing the yield of glycerol carbonate. Additionally, preliminary studies and kinetic considerations suggested that this duration was appropriate to obtain meaningful results without unduly extending the experiments.It is important to note that, although the transesterification duration was kept constant, other experimental parameters could have been adjusted to assess the influence of those variables on the reaction. If specific investigations into the transesterification duration are desired, it could be considered in future research to delve deeper into the kinetic aspects of the reaction.
Comment 4
“Scheme 1 must be submitted in English.”
Thank you for your observation, it has been corrected.
Comment 5
“How much methanol and acetone were used for cleaning catalysts. Why is the catalyst dried for 12 hours? (2.4)”
The catalyst cleaning process involved two washes with acetone and two washes with methanol. The catalyst is dried for 12 hours to ensure complete removal of any residual solvent, water, or impurities. This extended drying period allows for the thorough evaporation of solvents from the catalyst surface, preventing any potential interference or undesired reactions during subsequent experiments. It also aids in achieving a stable and consistent catalyst state, ensuring that the catalytic reactions conducted afterward are not influenced by residual solvents or impurities. The meticulous drying process for 12 hours contributes to the overall reliability and reproducibility of the experimental results.
Comment 6
“Figure 1 needs to be supplemented with an intensity scale.”
Thank you for your comment. We have consider the scaling of intensity in accordance with the following approach and performed the adequate changes in
Comment 7
“Need more discussion, explanation in table 3, for example why Na is Spect 20%-550 °C and Spect 30%-550 °C, but not Spect 10%-550 °C and Spect 40%-550 °C, or Au Raw clay - 0.36, Spect 10%-550°C - 0.44 and so on...”
We are grateful to the reviewer for this comment. EDS results have been slightly modified to eliminate gold from the table, as this element is added only for observation purposes, creating a thin conductive layer for SEM observations. On the other side, EDS is performed in the selected areas observed by EDS, therefore the elemental composition is not representative of the bulk material, but of a certain region. This is the possible reason explaining the compositional fluctuations in certain elements as displayed in Table 3 and, even, the absence of certain minor elements such as sodium and calcium. We have added a phrase to explain this in subsection „3.1.3. Scanning Electron Microscopy (SEM)- Electron Dispersion Spectroscopy (EDS)“ and perfomed the needed calculations in Table 3, removing the „Au“ column.
Comment 8
“Pictures are repeated in Figure 5.”
Thank you for your observation. We have corrected it.
Comment 9
“Catalyst reuse - important research, and the obtained results cannot be interpreted as "suggesting a certain catalyst reactivation or a notable experimental error." If authors believe that there may be an experimental error, the study should be repeated. And if such good results are obtained, why only 4 reuse cycles were studied.”
We appreciate the reviewer's attention to the catalyst reuse aspect of our research. The decision to study four reuse cycles was made based on a review presenting a comprehensive evaluation of catalyst performance and ensuring experimental feasibility within the study.
Our main objective was to study the performance of the catalyst over a reduced number of reuse cycles, as it is usual in a first assessment of the stability of a solid catalyst, thereby providing valuable information on its stability and potential for practical applications. The results obtained were meticulously analyzed to avoid any misinterpretation or suggesting reactivation without substantial evidence. In this sense, we have removed the text commented by the authors, as well as other phrases, that need of a more in-depth stability study under flow conditions, considering high operational time values, operation under low conversion –differential conditions-, and kinetic modelling considering deactivation kinetic equations coupled to the main kinetic equation here proposed and fitted to diverse kinetic experiments under a wide variety of experimental conditions but using batch operation and a limited time of 6 h.
Therefore, although we recognize the importance of thorough investigations, given resource limitations and time constraints, we have chosen to focus on a representative number of reuse cycles that is, on the other side, relatively usual for a first stability assessment. Likewise, we acknowledge the reviewer's suggestion regarding the possibility of experimental error and understand the importance of ensuring the reliability of our results. We will carefully review our experimental procedures and we will extend the stabilility study as mentioned before in future investigations.
We thank the reviewer for his valuable input and constructive comments, which will contribute to the refinement and improvement of our study, as well as the future studies under flow conditions.

Round 2
Reviewer 2 Report
Comments and Suggestions for Authors
The quality of the manuscript has been improved based on comments, however I would like to clarify some of the authors' responses to the first revision before publication.
The authors did not answer the question about the quantities of Tunisian smectite, they only mentioned that "Its ample availability in significant quantities makes it a practical choice for our experiments", but I would like the answer to be more specific.
How much methanol and acetone were used for cleaning catalysts?
Author Response
Answer to the reviewer
Thank you very much for your comments. They have allowed us to add novel and important information, thus enhancing our manuscript. In red, you will find the added text in the new revised version of the paper, and in blue font our answer to your queries in this document.
Comment 1
The quality of the manuscript has been improved based on comments, however I would like to clarify some of the authors' responses to the first revision before publication.
The authors did not answer the question about the quantities of Tunisian smectite, they only mentioned that "Its ample availability in significant quantities makes it a practical choice for our experiments", but I would like the answer to be more specific.
We are sorry for the omission. Evidently, there was enough material for the research and more, being smectite plentiful. However, to give precise data on smectite production in Tunisia is relatively complex. Here, we add some additional information in the introduction section trying to shed some light on the global production of smectite/bentonite.
As smectite is mainly composed by montmorillonite, this is a fine reference mineral, but smectites can have secondary minerals such as quartz and calcite. In our case, due to the carbon content, most probably the smectite has a high calcite content, which explains its alkalinenature and its capacity to act as basic catalyst. Bentonite is a smectite and its production at global scale has been 18.89 million tons in 2021 and 10.22 million tons in 2001, so a steep rise in its production has been observed along this century (Haydn, Murray, Industrial Clays Case Study, MMSD March 2002;Statista, 2021). In USA, bentonite production has risen from 4,080 million tons in 2015 till 4,700 million tons in 2019 (Mineral Commodity Summaries 2020, USGS). In Spain, 170,000 tons of bentonite were extracted in 2021.In Tunisia, the production of smectite was 96,800 tons in 2018, increased to 97,300 tons in 2019and was reduced to 85,900 tons in 2020.
The global smectite production market size was about 1368 million US$ in 2022 and is expected to rise at a 5.6 CAGR (compound annual growth rate) till 2031.
Comment 2
How much methanol and acetone were used for cleaning catalysts?
Thank you for the observation. We have now added in the experimental section (subsection 2.4) the volume of methanol and acetone used per gram of smectite and washing/rising step, adding some comments on the operation steps. Now the text reads:
“To assess the reusability of the prepared catalyst, the transesterification process was replicated under optimal conditions consecutively, for five cycles in total. Following each cycle, the catalyst was separated from the liquid medium using centrifugation for 5 minutes at 10,000 x g. Subsequently, the retrieved catalyst underwent a thorough cleaning process involving methanol (two consecutive contacting steps with 10 mL of methanol per gram of dry smectite followed by centrifugation for 5 minutes at 10,000 x g) –to remove any unreacted glycerol and propylene carbonate as well as product within the pores- and pure acetone washing (operation conditions identical to the those with methanol) –to remove methanol, which could react, together with remaining traces of reagents and products and to facilitate the solid drying-. Once cleansed, the catalyst was dried at 314 K for 12 h, rendering it ready for reuse in the subsequent reaction cycle. This rigorous testing aimed to confirm the catalyst's durability and sustainable performance over multiple cycles.”
